# A probabilistic model for the ultradian timing of REM sleep in mice

**Sung-Ho Park**, **Justin Baik**, **Jiso Hong**, **Hanna Antila**, **Benjamin Kurland**, **Shinjae Chung**, **Franz Weber** *

Department of Neuroscience, Perelman School of Medicine, Chronobiology and Sleep Institute, University of Pennsylvania, Philadelphia, Pennsylvania, United States of America

* fweber@pennmedicine.upenn.edu

**Data Availability Statement:** All data files are available for download at this URL: https://upenn.box.com/s/3zcesr4a7l7hgb9andmq4di4t6zvaoql.

**Funding:** This work was supported by the National Institutes of Health (R01HL149133 to FW,

## Abstract

A salient feature of mammalian sleep is the alternation between rapid eye movement (REM) and non-REM (NREM) sleep. However, how these two sleep stages influence each other and thereby regulate the timing of REM sleep episodes is still largely unresolved. Here, we developed a statistical model that specifies the relationship between REM and subsequent NREM sleep to quantify how REM sleep affects the following NREM sleep duration and its electrophysiological features in mice. We show that a lognormal mixture model well describes how the preceding REM sleep duration influences the amount of NREM sleep till the next REM sleep episode. The model supports the existence of two different types of sleep cycles: Short cycles form closely interspaced sequences of REM sleep episodes, whereas during long cycles, REM sleep is first followed by an interval of NREM sleep during which transitions to REM sleep are extremely unlikely. This refractory period is characterized by low power in the theta and sigma range of the electroencephalogram (EEG), low spindle rate and frequent microarousals, and its duration proportionally increases with the preceding REM sleep duration. Using our model, we estimated the propensity for REM sleep at the transition from NREM to REM sleep and found that entering REM sleep with higher propensity resulted in longer REM sleep episodes with reduced EEG power. Compared with the light phase, the buildup of REM sleep propensity was slower during the dark phase. Our data-driven modeling approach uncovered basic principles underlying the timing and duration of REM sleep episodes in mice and provides a flexible framework to describe the ultradian regulation of REM sleep in health and disease.

## Author summary

During sleep, the mammalian brain repeatedly alternates between two brain states: REM and NREM sleep. This ultradian oscillation constitutes a fundamental brain rhythm, the so-called sleep cycle, which is conserved across mammalian species. However, the mechanisms that generate the sleep cycle are still largely unknown. A conserved statistical feature of mammalian sleep is that the durations of REM sleep and subsequent NREM sleep are positively correlated. This correlation suggests that REM sleep impacts the amount of the

R01NS110865 to SC), by the Brain & Behavior Research Foundation (NARSAD Young Investigator grant #27799 to FW), by the Margaret Q. Landenberger Research Foundation (to FW) and by a fellowship from the Sigrid Juselius Foundation (https://www.sigridjuselius.fi) awarded to HA. The funders had no role in study design, data collection and analysis, decision to publish, or preparation of the manuscript.

**Competing interests:** The authors have declared that no competing interests exist.

following NREM sleep and thereby influences its own timing. Here, we developed a statistical model that accurately describes the relationship between the preceding REM and following NREM sleep duration during spontaneous sleep in mice. We applied this model to investigate the relationship between REM sleep and the quality of future NREM sleep, and to uncover factors that determine the timing and duration of REM sleep episodes. Using our model-based approach, we identified three major factors shaping the ultradian regulation of REM sleep: Two types of sleep cycles, a period of light NREM sleep during which transitions to REM sleep are suppressed, and a propensity that influences the subsequent REM sleep duration.

## Introduction

During sleep, the mammalian brain alternates between two distinct states—rapid eye movement (REM) sleep and non-REM (NREM) sleep. The cyclic occurrence of REM sleep constitutes the REM-NREM or sleep cycle, an ultradian rhythm on a minute-to-hour time scale shared by mammals [1], birds [2], and reptiles [3,4]. Although we know in great detail about the neural mechanisms underlying oscillations on a millisecond-to-second timescale [5] or about transcriptional/translational oscillators generating circadian rhythms [6], we lack knowledge of how the brain generates ultradian rhythms.

Transitions from NREM to REM sleep are thought to be controlled by a network of interconnected REM sleep-promoting (REM-on) and REM sleep-suppressing (REM-off) neurons [7–10]. Research in the last decade has identified key populations of REM-on and REM-off neurons in the brainstem and hypothalamus that powerfully promote or suppress REM sleep and has mapped their connectivity at unprecedented detail [11–18]. However, the mechanisms that regulate when the brain state transitions from NREM to REM sleep and thereby determine the duration of the sleep cycle are still largely unknown.

A common statistical feature of mammalian sleep, observed in multiple species including humans is that the duration of REM sleep is positively correlated with the time till the next REM sleep period (inter-REM interval) [19–25]. This correlation is thought to be the manifestation of a homeostatic process that operates on the ultradian time scale [19,20]: A propensity for REM sleep builds up during the inter-REM interval and partially discharges during REM sleep. Longer REM periods, therefore, lead to a stronger discharge of the REM propensity and thus require longer intervals for accumulation to re-enter REM sleep. According to this model, the sleep cycle is not generated by an oscillator circuit, as originally proposed [26], but rather is the consequence of an hourglass-type ultradian process [19,20,27,28], the neural or molecular correlates of which are still largely unknown.

Recently, the hourglass process has been proposed to be the result of a refractory period following REM sleep, during which transitions to REM sleep are effectively suppressed [29–31]. The probability of NREM to REM sleep transitions is reduced particularly after long REM sleep periods (>1 min) in rats [32], which may explain the positive correlation between the duration of REM sleep and the inter-REM interval. However, although possibly being a crucial subunit within the sleep cycle, there is still no quantitative definition of the refractory period, nor is there an understanding of how the quality of NREM sleep and its microarchitecture may be changed during and after the refractory period. Furthermore, as the duration of REM sleep influences the duration of the following inter-REM interval, it is crucial for our understanding of the ultradian sleep cycle to identify factors that regulate the duration of REM sleep episodes.

The sleep pattern in mammals is further complicated by sequences of REM sleep episodes, whose timing deviates from the statistics expected from an hourglass process. In rats, such sequences comprise several temporally close (< 3 min) REM sleep periods [33–35] and their presence results in a bimodal distribution of the inter-REM interval durations. In addition to the rat, REM sleep sequences have been reported in multiple mammalian species including humans [21,36,37], suggesting that they are a conserved phenomenon in mammalian sleep. An increased frequency of REM sleep sequences underlies the homeostatic rebound following a loss in REM sleep induced by exposure to cold temperatures [33,34] and is characteristic of sleep in stressed animals [38,39]. However, the mechanisms underlying the induction of REM sleep sequences and their defining electrophysiological properties are still largely unclear.

Here, we developed a conditional Gaussian mixture model (GMM) that specifies the relationship between the preceding REM sleep duration and the subsequent time spent in NREM sleep. We applied this model to systematically separate short cycles, which form sequences of REM sleep, from long cycles, which are characterized by the positive correlation between REM and subsequent NREM sleep. For long cycles, we defined the duration of the refractory period as a function of the preceding REM sleep duration. Next, we analyzed the EEG and other features of NREM sleep to identify defining properties of NREM sleep during the refractory period. We then used the cumulative distribution function (CDF) of the model as a measure for REM sleep propensity and found that entering REM sleep at higher propensity resulted, on average, in longer REM sleep episodes and reduced the EEG power. Finally, we employed the model to uncover changes in the regulation of REM sleep between the light and dark phase. Altogether, our model-based approach uncovered basic principles underlying the timing and duration of REM sleep episodes.

## Results

### A probabilistic model relating NREM sleep to preceding REM sleep

Consistent with the terminology introduced in earlier studies, we refer to the time interval between two successive REM periods as the inter-REM interval, while a sleep cycle (also called REM-NREM cycle) comprises one REM episode and the following inter-REM interval (**Fig 1A**) [21,40]. We recorded spontaneous sleep in wildtype mice during the light phase and confirmed in our data set the positive correlation between the preceding REM sleep duration (REM$_{pre}$) and the subsequent inter-REM interval (**Fig 1B left and S1 Table**). For further analysis, we divided the inter-REM interval into its total duration of NREM sleep, |N|, and total duration of wakefulness, |W| (**Fig 1A**). Consistent with previous studies [19,20,25], we found that in mice, as in rats, REM$_{pre}$ was more strongly correlated with |N| than with either the total inter-REM interval or |W| (**Fig 1B and S1 Table**).

To systematically describe the interaction between REM sleep and the duration and quality of subsequent NREM sleep, we developed a probabilistic model that specifies the relationship between REM$_{pre}$ and |N|. Applying the natural logarithm (ln) to |N| and plotting the distribution of ln(|N|) separately for increasing 30 s bins of REM$_{pre}$, we found that for values of REM$_{pre}$ in the range from 30 s to 150 s, ln(|N|) forms a bimodal distribution, reflecting the presence of short and long sleep cycles in the hypnogram (**Fig 2A and 2B**). For REM$_{pre} \geq 150$ s, the distribution became unimodal. The distribution of ln(|N|) appeared to be a mixture of two bell-shaped distributions, with the relative weights of each distribution depending on REM$_{pre}$. Based on this observation, we estimated for each 30 s bin of REM$_{pre}$ the distribution of ln(|N|) using a two component GMM (**Methods**). More precisely, for each REM$_{pre}$ bin, we modeled the distribution of ln(|N|) as the weighted sum of two Gaussians, $k_{short} \cdot N(\mu_{short}, \sigma^2_{short})$ $+ k_{long} \cdot N(\mu_{long}, \sigma^2_{long})$, where k, μ, and σ refer to the weighting factor, mean, and standard

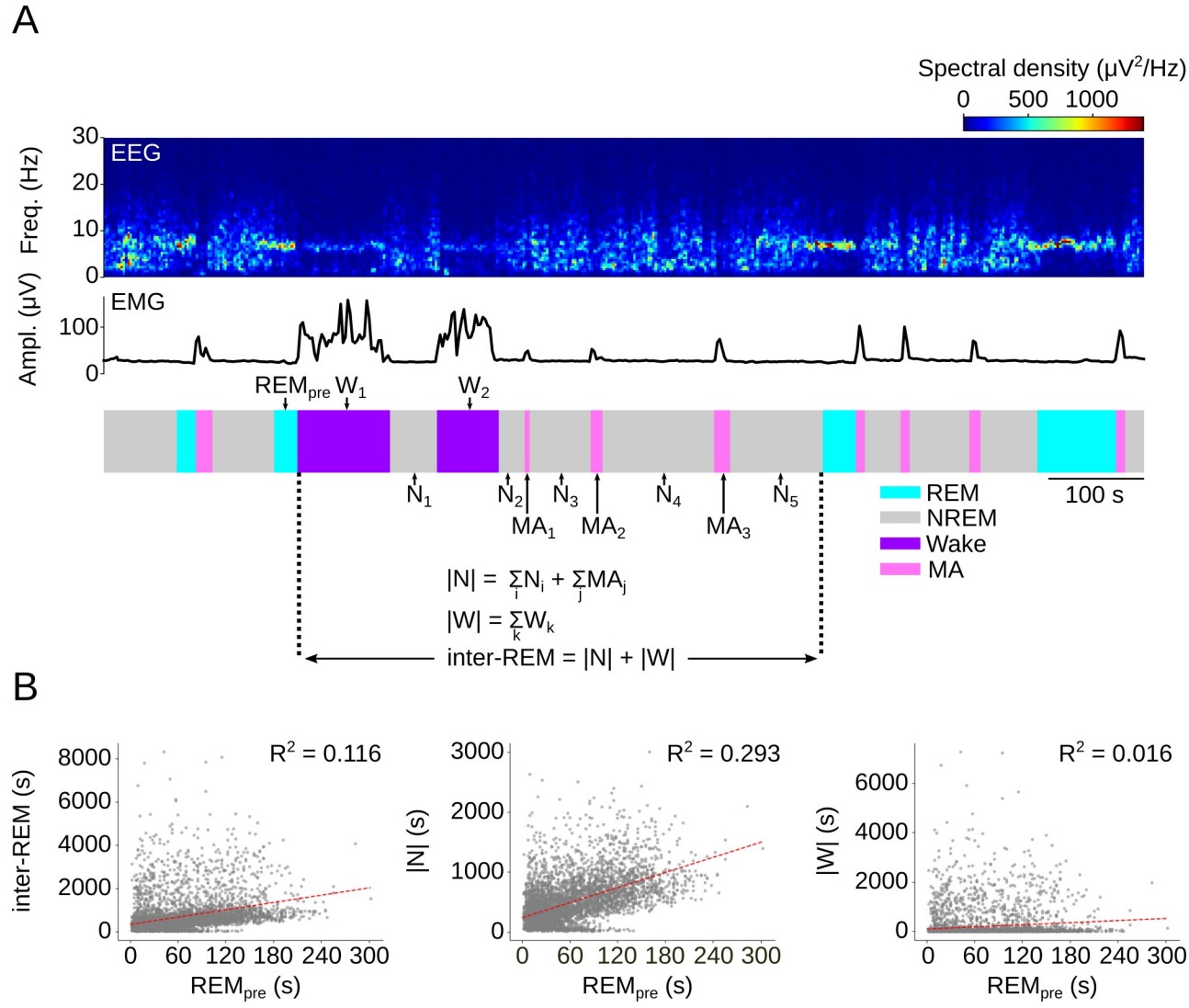

**Fig 1. Correlation between REM sleep duration and inter-REM interval.** (A) Example EEG power spectrogram, EMG amplitude, and hypnogram with definitions of terms. $REM_{pre}$, duration of the preceding REM sleep episode; inter-REM, duration of subsequent interval till next REM episode; |W|, sum of the durations of all wake episodes during the inter-REM interval; MA, microarousal (wake bouts $\leq$ 20 s); |N|, sum of the durations of all NREM episodes (including MAs) during the inter-REM interval. (B) Scatter plots with $REM_{pre}$ on the x-axis and subsequent inter-REM duration (left), |N| (middle), and |W| (right) on the y-axis. The results of linear regression fits are shown in red (P<0.00001 for all 3 slopes, n = 5098 sleep cycles from 72 mice recorded during the light phase).

deviation of each Gaussian, respectively (**Fig 2C**). To test whether the GMM is a valid model for the data, we performed the Lilliefors-corrected Kolmogorov-Smirnov test for each 30 s bin and did not find sufficient evidence to reject the null hypothesis that the data are indeed drawn from the estimated Gaussian mixture distributions (**S2 Table and Methods**). This was true regardless of the specific threshold used to score microarousals (MAs) (**S3 Table**).

We found that the weight of the long Gaussian distribution, $k_{long}$, steadily increased with $REM_{pre}$, indicating that the probability for long cycles was larger the longer the preceding REM episode (**Fig 2D**). Second, the mean of the long Gaussian distribution, $\mu_{long}$, increased with $REM_{pre}$, while the standard deviation, $\sigma_{long}$, decreased. The mean and standard deviation of the short cycles, $\mu_{short}$ and $\sigma_{short}$, both decreased with larger $REM_{pre}$ values.

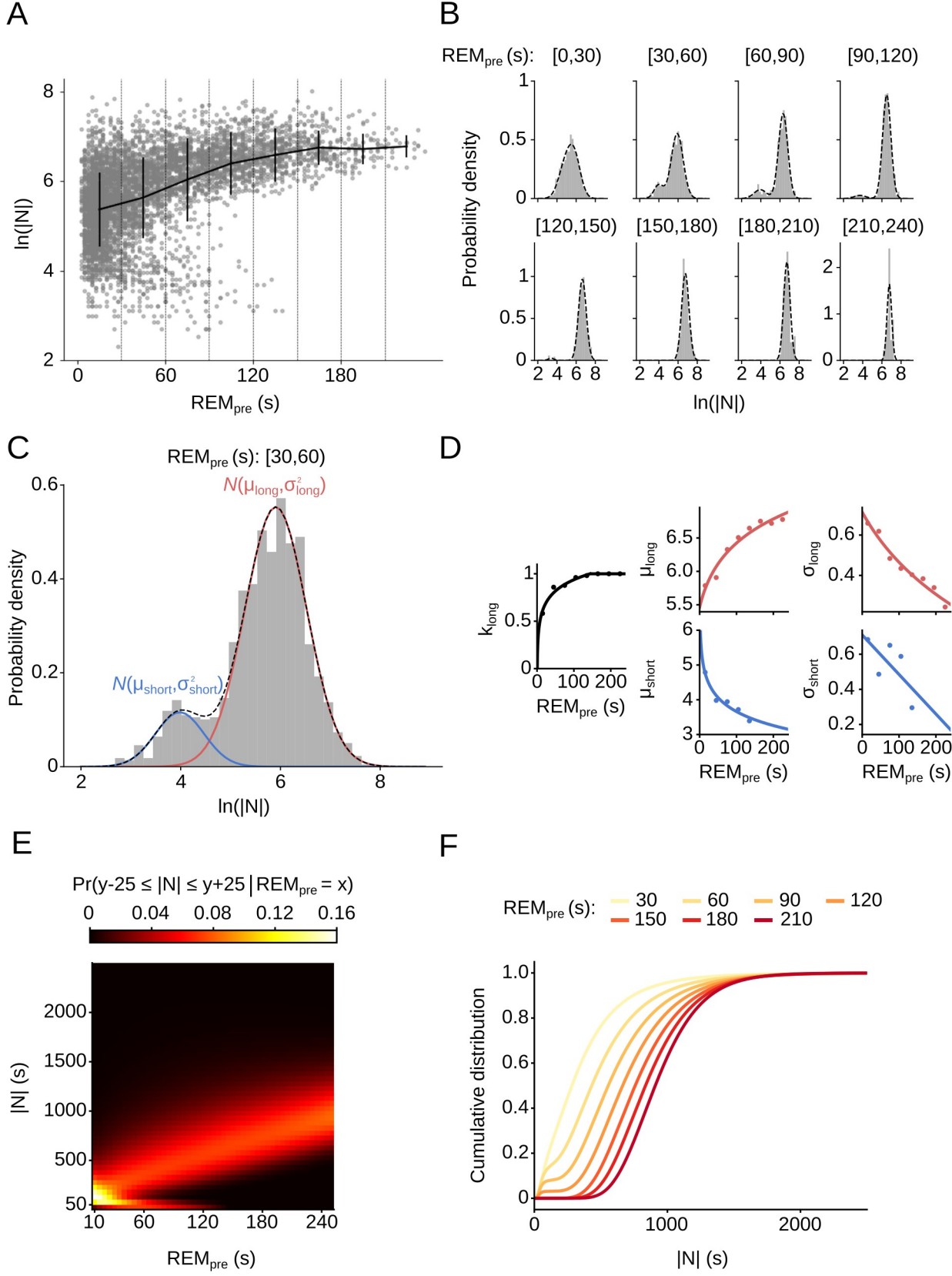

**Fig 2. Conditional GMM to describe the relationship between REM$_{pre}$ and subsequent NREM.** (A) Scatter plot of REM$_{pre}$ vs. ln(|N|). Vertical dashed lines indicate consecutive 30 s bins of REM$_{pre}$. Solid black lines represent the mean and standard deviation of ln(|N|) for each 30 s bin. (B) Histograms and probability density plots of ln(|N|) for consecutive REM$_{pre}$ bins as indicated on top. Probability densities were computed using a GMM. The notation [a, b) refers to the bin a ≤ REM$_{pre}$ < b. (C) Histogram of ln(|N|) for inter-REM intervals preceded by REM episodes in the range 30 s ≤ REM$_{pre}$ < 60 s. A GMM composed of two Gaussian distributions captures well the bimodal distribution of ln(|N|). The mean and standard deviation of the Gaussian for long and short cycles are referred to as μ$_{long}$, σ$_{long}$, and μ$_{short}$, σ$_{short}$, respectively. (D) Estimates of GMM parameters as a function of REM$_{pre}$. The mixture parameter, k$_{long}$, denotes the probability that a sleep cycle belongs to the long Gaussian distribution. For each parameter, we fitted a linear or logarithmic function describing its dependence on REM$_{pre}$. (E) Heatmap in which each grid cell (x,y) represents the probability of transitioning from NREM to REM in between |N| - 25 ≤ y ≤ |N| + 25 s following a REM episode of duration REM$_{pre}$ = x s for x in [10, 15, . . ., 250]. Each column of the heatmap sums up to 1. (F) Cumulative distribution function (CDF) of the GMM for 7 different values of REM$_{pre}$. Each line represents, for the given REM$_{pre}$ value, the likelihood of entering the next REM period within |N| s of NREM sleep since the preceding REM episode.

Next, to describe the relationship between REM$_{pre}$ and each of the GMM parameters, we fit either linear or logarithmic functions to the parameter estimates (**Figs 2D and S1A and S4 Table**). By finding a function that captures the relationship between REM$_{pre}$ and each Gaussian mixture parameter, we were able to explain the amount of subsequent NREM sleep conditional on REM$_{pre}$ using a single probability model (see **Methods**).

We visualized the complete probability model using a heatmap (**Fig 2E**). Each grid cell (x,y) on the heatmap color-codes the probability to transition into REM sleep after |N| = *y* s of NREM sleep since the last REM sleep period of duration REM$_{pre}$ = *x* s. As expected from the distribution of ln(|N|) (**Fig 2A and 2B**), we observed two modes along |N|. The lower mode comprises only short inter-REM intervals and exists only for REM$_{pre}$ < 150 s. The second mode contains longer inter-REM intervals with larger |N| values. We refer to sleep cycles with |N| in the lower mode as sequential, as they result in a sequence (or cluster) of closely inter-spaced REM sleep periods. Sleep cycles that are part of the long distribution are termed single cycles. One characteristic of single cycles is that the mean values of |N| continuously increase with REM$_{pre}$, i.e. longer REM sleep episodes are followed by larger amounts of NREM sleep before re-entering REM sleep. As further analyzed below, another key characteristic of single cycles is that the preceding REM period is followed by an interval of NREM sleep during which it is extremely unlikely to transition to REM sleep; the duration of this refractory period increases with REM$_{pre}$.

Next, we computed the CDF of the conditional GMM for different values of REM$_{pre}$ (**Fig 2F**). Each CDF represents the probability that the animal transitions to REM sleep within |N| s of NREM sleep. In general, the longer the preceding REM period, the more NREM sleep is required to reach the same cumulative probability of re-entering REM sleep. But, irrespective of REM$_{pre}$, within 2000 s of NREM sleep, the animal will most likely (> 99.46%) transition to the next REM period. For short REM$_{pre}$ values, the CDF immediately rises because of the comparably high probability for the occurrence of sequential cycles. In contrast, for long REM sleep episodes (REM$_{pre}$ ≥ 150 s), the CDF initially stays close to zero and only starts rising once it leaves the refractory period. Altogether, our statistical model to describe the relationship between REM$_{pre}$ and |N| suggests the existence of two different types of sleep cycles: Sequential cycles form sequences of REM sleep episodes, whereas single cycles are characterized by the existence of a refractory period, the duration of which increases with the duration of the preceding REM episode.

A previous study accounted for the presence of sequential sleep cycles and the resulting bimodality of the inter-REM distribution by using a mixture of two Poisson distributions [35]. Consistent with our findings, the authors reported that the mean duration of the inter-REM interval and the probability of observing a long inter-REM interval both increased with REM$_{pre}$. However, a key property of the Poisson distribution is that the variance increases with the mean, which is not the case for our dataset for both the total inter-REM duration and |N|

(S2 Fig). Our conditional GMM instead reproduces the reduction in the variance for increasing values of $REM_{pre}$.

## Sequential vs single cycles

Using our model, we defined a data-driven criterion to separate single from sequential cycles (**Fig 3A left**). For each 2.5 s increment of $REM_{pre}$, we calculated the probability density functions (PDFs) of both the short and long Gaussian distribution. The intersection point of the two distributions optimally separates single from sequential cycles (**Fig 3A right and Methods**). In total, 19.3% of all cycles were sequential (**Fig 3B**). Since $k_{long} = 1$ for $REM_{pre} \geq 150$ s, sequential cycles exist only for $REM_{pre} < 150$ s. In contrast, single cycles exist for the entire range of $REM_{pre}$, and consequently, their $REM_{pre}$ values are on average larger than those of sequential cycles (**Fig 3C**). The total duration of NREM sleep during sequential cycles reached values up to 222.5 with a mean of 84.96 s ± 40.92 s (mean ± s.d.) (**Fig 3D**). In more than 50% of cases, REM sleep sequences comprised only two REM sleep episodes, forming one cycle; in rare cases, they included up to 5 consecutive cycles (**Fig 3E**).

To test for further differences between sequential and single cycles, we examined the EEGs for both types of cycles (**Fig 3F**). For both the parietal and prefrontal EEG, the general shape of the spectral density during REM sleep was similar, although there were differences in the overall power, which were particularly pronounced for the prefrontal EEG. Further analysis, however, showed that these differences were the result of the differences in the REM episode duration, $REM_{pre}$, between single and sequential cycles (**Fig 3C**). We determined the spectral densities for different REM sleep durations (**S3A Fig**) and used these to calculate weighted averages based on the distribution of $REM_{pre}$ for single and sequential cycles, respectively (see **Methods**). The weighted averages were very similar to the actual spectral densities, suggesting that the power differences in the REM sleep EEG between single and sequential cycles are the result of the differences in $REM_{pre}$ (**S3B Fig**).

In contrast to REM sleep, the spectral density for NREM sleep exhibited considerable differences between sequential and single cycles, particularly in the parietal EEG (**Fig 3F**). Compared with NREM sleep during single cycles, the NREM δ power (0.5–4.5 Hz) during sequential cycles was strongly reduced, while the θ power (5–9.5 Hz) was enhanced. We observed very similar changes in the δ and θ power when varying the threshold used to score MAs (**S4B Fig**). Thus, the EEG displayed two features resembling the EEG during REM sleep (reduced δ and increased θ power), suggesting that NREM sleep during sequential cycles may constitute an intermediate state between NREM and REM sleep [41].

## Refractory vs. permissive periods during single cycles

During the inter-REM interval of single cycles, REM sleep is followed by a refractory period during which the probability of NREM to REM transitions is close to zero (**Fig 2**). Using the conditional GMM, we formulated a data-driven definition of the refractory period (**Fig 4A left**). For each 2.5 s increment of $REM_{pre}$, we calculated the 1st percentile of the long Gaussian distribution and set it as the threshold separating the refractory period from the remaining permissive period (**Fig 4A right**). Notably, the duration of the refractory period is approximately twice the duration of $REM_{pre}$ (**Fig 4B**).

Next, we tested whether the EEG and other properties of NREM sleep, such as sleep spindles and microarousals (MAs), differ between the refractory period and the following permissive period that ranges from the end of the refractory period till the onset of the next REM episode. We computed the spectral density of the EEG during NREM sleep and found that the δ, θ, and σ (10–15 Hz) power of both the prefrontal and parietal EEG were lower during the

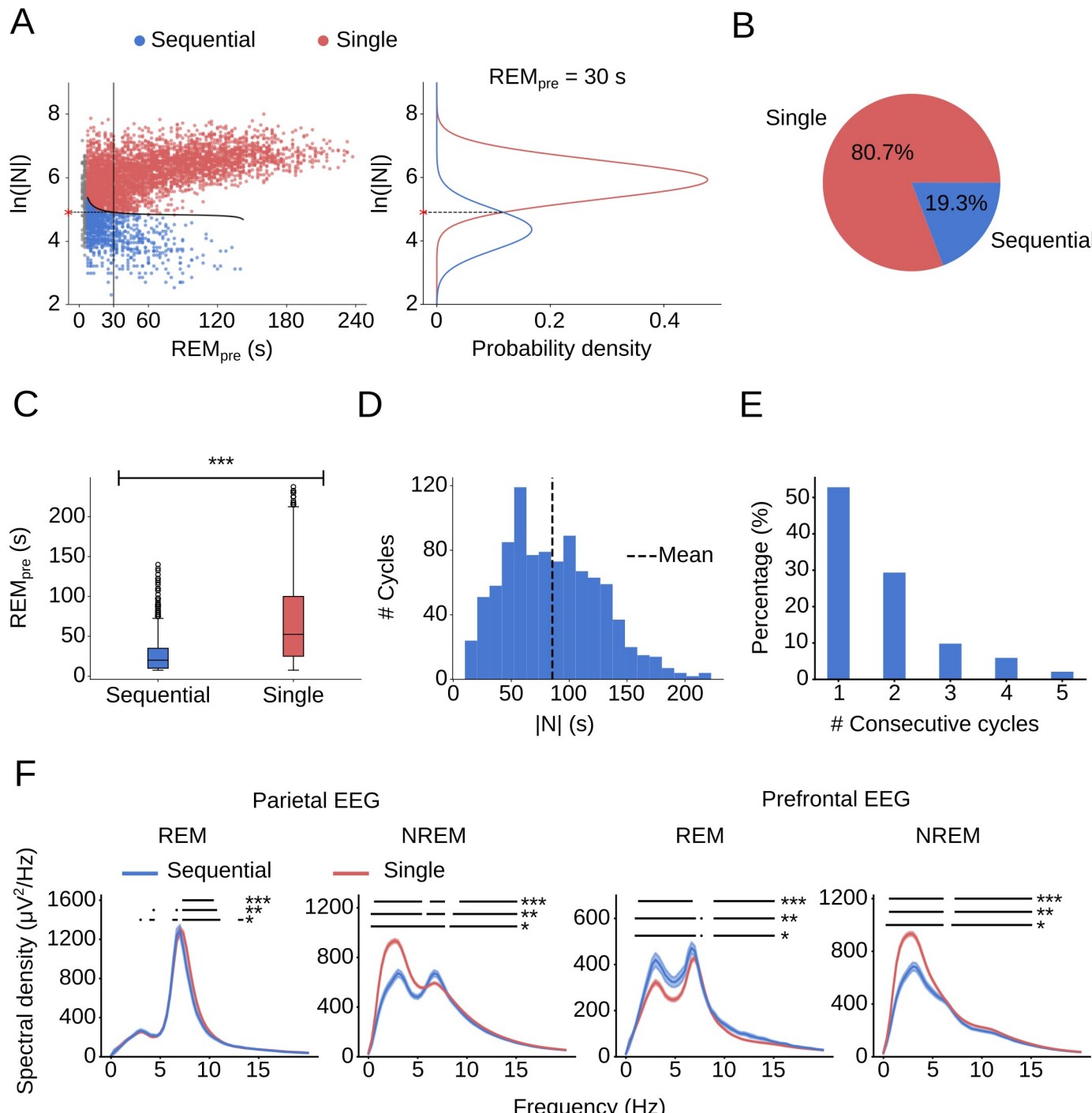

**Fig 3. Sequential vs single cycles.** (A) (Left) Scatter plot of $REM_{pre}$ vs. $ln(|N|)$ with color-coded single and sequential cycles. The threshold optimally separating sequential from single cycles is shown in black. (Right) Illustration of how the threshold was determined for $REM_{pre} = 30$ s. The probability density functions (PDFs) for the two distributions of the GMM (for $REM_{pre} = 30$ s) are plotted along the y-axis. The red asterisk indicates the value of $ln(|N|)$ at which the two Gaussians intersect. Values of $ln(|N|)$ below the intersection point are more likely to be drawn from the short distribution and are consequently labeled as sequential cycles. Gray points correspond to cycles with $REM_{pre} < 7.5$ s for which the conditional GMM is not defined (**S1B Fig and Methods**). (B) Pie chart indicating the percentage of single and sequential cycles. (C) Box plot comparing $REM_{pre}$ for single and sequential cycles. For sequential cycles, $REM_{pre}$ was shorter than for single cycles (Welch's t-test, t = -35.13, p = 2.59e-228, $n_{sequential} = 947$, $n_{single} = 3961$). (D) Histogram of $|N|$ for sequential cycles. The vertical dashed line indicates the mean (85.46 s ± 40.92 s; mean ± s.d.). (E) Bar plot showing the percentage of the number of cycles within a REM sleep sequence. Over half of REM sleep sequences contain only one cycle (i.e. comprise two REM periods). (F) Spectral density of parietal ($n_{sequential} = 947$, $n_{single}, = 3961$) and prefrontal ($n_{sequential} = 936$, $n_{single} = 3919$) EEG during REM and NREM sleep for both sequential and single cycles. Horizontal lines indicate frequencies at which the spectral density of

sequential and single cycles are statistically different at various σ levels; (Welch's t-test, * p<0.05; ** p<0.01; *** p<0.001). One recording did not contain a prefrontal EEG channel. Shadings, 99% confidence interval (CI).

refractory period (**Figs 4C and S5A**). In addition to these differences in the EEG, the spindle rate was reduced during the refractory period while MAs were more frequent (**Fig 4D and 4E**). Sleep spindles have recently been implicated in promoting REM sleep and their overall increased rate during the permissive period may thus facilitate transitions to REM sleep [42].

We then analyzed the time course of the different prefrontal EEG power bands and the rate of MAs and spindles throughout the sleep cycle by normalizing the durations of both the refractory and permissive period and dividing them into quarters. The θ power, σ power, and rate of sleep spindles were strongly reduced after REM sleep, increased with downward concavity throughout the refractory period, reached a plateau near its end, and then continued increasing with upward concavity throughout the permissive period (**Fig 4F**). The overall time course of the θ and σ power did not depend on the exact value of the threshold used to score MAs (**S4C Fig**). The increase of these values in the final quarter of the permissive period reflects the transition stage between NREM and REM sleep, which is characterized by increased spindle activity, increased θ and reduced δ power (**S5C Fig**) [41]. Importantly, the normalized time course of the θ power, σ power, and spindle rate was consistent regardless of $REM_{pre}$ (**Fig 4F**), suggesting that, as the duration of the refractory period increased with longer REM periods, the rate at which the θ power, σ power, and frequency of spindles increased was proportionally reduced. Plotting these quantities throughout the refractory period along a non-normalized time axis, we indeed found a slower rise in their time courses following longer REM periods (**Figs 4G and S5D**). The rate of MAs followed the opposite pattern. It was strongly increased after REM sleep and decreased with upward concavity throughout the refractory period before reaching a plateau, and then continued to decay with downward concavity throughout the permissive period (**Fig 4F**). We found a similar pattern, irrespective of the used MA threshold (**S4C Fig**). The time course at which the MA rate declined also depended on $REM_{pre}$: The longer $REM_{pre}$, the less steep the decay (**S5D Fig**). The fact that the inflection point of the time courses of the θ power, σ power, spindle and MA rate all occur at the threshold suggests that our data-driven definition of the refractory and permissive period reflects a natural separation within single sleep cycles.

In contrast, the time course of the δ power was not aligned with the threshold between the refractory and permissive period (**S5B Fig**). Although its general time course, with an increase at the beginning of the inter-REM interval followed by a decrease, was consistent for all ranges of $REM_{pre}$, the normalized time bin at which the δ power started decaying varied with $REM_{pre}$ and was not systematically related with the threshold between the refractory and permissive period. Thus, although the time course of the δ power and, in particular, its overall power was strongly influenced by $REM_{pre}$, consistent with a previous study [14], it did not reflect the probability of NREM to REM sleep transitions.

Finally, to test whether the observed changes in the EEG power and sleep microarchitecture during the refractory period were specifically due to REM sleep, we compared NREM sleep after REM sleep with NREM sleep following wake periods of equal duration (**Figs 4G and S5D and S5E**). In general, the power of all tested EEG bands was less strongly modulated after a period of wakefulness. After REM sleep, the θ and σ power showed a stronger reduction and a steeper increase (**Fig 4G**) compared to their time courses following wake periods of equal duration (**S5E Fig**). The rate of MAs was also more strongly modulated following bouts of REM sleep (**S5D and S5E Fig**). Of all tested variables, the θ power and rate of MAs were the least affected by preceding wakefulness (**S5E Fig**). Thus, the observed changes in the NREM

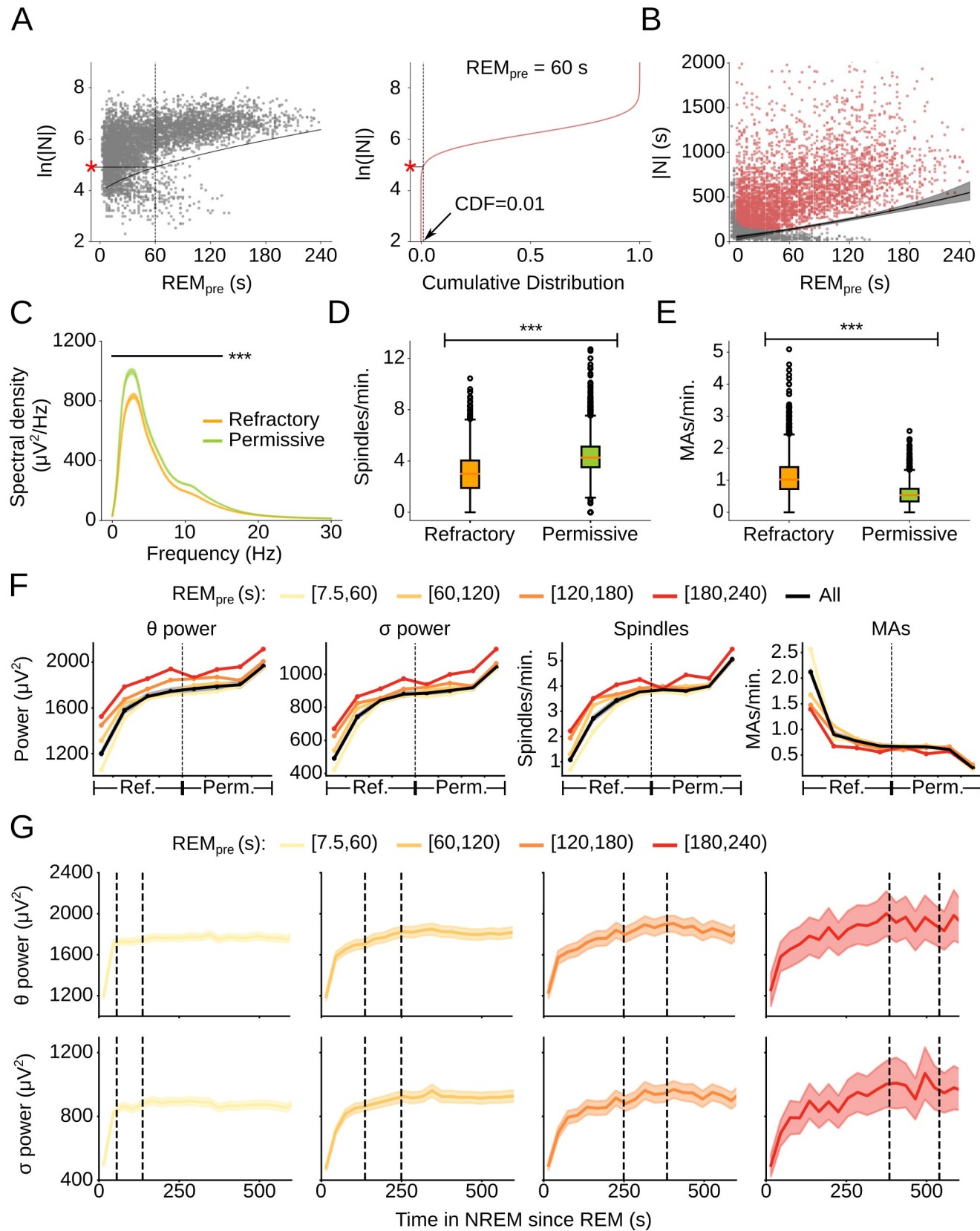

**Fig 4. Refractory and permissive periods during single cycles.** (A) (Left) Scatter plot of $REM_{pre}$ vs. $\ln(|N|)$ along with boundary (solid line) separating the refractory from the permissive period within single cycles. (Right) Illustration of how the threshold separating the refractory from the permissive period was determined for $REM_{pre} = 60$ s. The CDF of the long Gaussian distribution is plotted along the y-axis. The value of $|N|$ for which $CDF(\ln|N|) = 0.01$ (indicated by the red asterisk) corresponds to the duration of the refractory period. (B) Scatter plot of $REM_{pre}$ vs. $|N|$ along with the threshold separating the refractory from the permissive period. Of note, the refractory period is only defined for single cycles (red dots). The black line represents the threshold and the shading indicates the 99% confidence interval (CI) from 10,000 bootstrap iterations. (C) Spectral density of the prefrontal EEG for NREM sleep during the refractory and permissive period. The densities for both periods are statistically different for frequencies in the range 0–15 Hz (Welch's t-test, *** $p < 0.001$, $n_{refractory} = n_{permissive} = 3892$). Shadings, 99% CI. (D) Box plot comparing the rate of sleep spindles during the refractory and permissive period (Welch's t-test, $t = -36.96$, $p = 0.0$, $n_{refractory} = n_{permissive} = 3908$). The rate was calculated as the number of spindles per 1 min of NREM sleep. (E) Box plot comparing the rate of MAs during the refractory and permissive period (Welch's t-test, $t = 50.32$, $p = 0.0$, $n_{refractory} = n_{permissive} = 3908$). The rate was calculated as the number of MAs per 1 min of NREM sleep. (F) Progression of θ power, σ power, spindle rate, and MA rate throughout the refractory and permissive period for different values of $REM_{pre}$. The refractory period is defined as outlined in A. The permissive period comprises the time from the end of the refractory period to the onset of the next REM period. The durations of both the refractory and permissive period were normalized to unit length and subdivided into quartiles of equal normalized duration. The average for all $REM_{pre}$ values is shown in black. Shadings, 99% CI. (G) Progression of θ power (Row 1) and σ power (Row 2) on non-normalized time scale during the first 600 s of NREM sleep during the inter-REM interval for different values of $REM_{pre}$. The two vertical dashed lines indicate the lowest and highest threshold separating the refractory from the permissive period corresponding to the low and high bound of $REM_{pre}$. Shadings, 99% CI.

EEG power and sleep microarchitecture, especially in the θ power and MA rate, were particularly associated with preceding REM sleep. These findings suggest that REM sleep has different consequences on the subsequent quality of NREM sleep than does wakefulness.

## Relationship between wakefulness and NREM sleep

In addition to $REM_{pre}$, $|N|$ may also be modulated by wake periods in the inter-REM interval. In our data set, only 12.9% of sequential cycles contained wake periods (**Fig 5A**). 43.0% of single cycles were not interrupted by wakefulness (**Fig 5B and 5C**) and 42.7% contained only one or two wake episodes (**Fig 5C**). In general, $|N|$ was larger for single cycles with larger $|W|$ (**Fig 5B and 5D**) and also for cycles with a larger number of wake episodes (**Fig 5D**). We observed similar relationships, irrespective of the threshold used to score MAs (**S6A and S6B Fig**). By comparing $|N|$ for cycles with different $|W|$ while keeping the number of wake episodes constant, and also comparing $|N|$ for cycles with different numbers of wake episodes while keeping $|W|$ constant, we confirmed that both $|W|$ and the number of wake episodes within an inter-REM interval contribute to longer $|N|$ (**S7 Fig**).

To test how wake episodes affect the EEG power during NREM sleep, we computed the θ and σ power before and after wake episodes during the inter-REM interval of single cycles. The power in both frequency bands increased during the preceding NREM episode, dropped to levels lower than those at wake onset, and then started rising again throughout the following NREM episode (**Fig 5E**; see **S6C Fig** for different MA thresholds). We observed the same pattern for the rate of spindles, whereas the rate of MAs was enhanced after wake and then decayed throughout NREM sleep. The drop in the spindle rate and the increase in the MA rate was larger the longer the intervening wake episode (**Figs 5F and S6C**). The drop in θ power, on the other hand, first increased and then declined with the duration of the intervening wake episodes. Thus, each wake episode leads to a reduction in quantities that reflect the probability of NREM to REM transitions (**Fig 4**), possibly resulting in more NREM sleep, which could explain the positive correlation of $|N|$ with the total duration and number of wake episodes.

## Correlation between model CDF and REM sleep duration

Next, we aimed to determine factors influencing the duration of REM sleep. First, we tested whether the REM duration ($REM_{post}$) is influenced by the preceding REM duration ($REM_{pre}$). We found that $REM_{post}$ is negatively correlated with $REM_{pre}$ (slope = -0.085, $R^2 = 0.0070$, $p = 7.17e-09$). The slope of the negative correlation was larger for sequential cycles (slope =

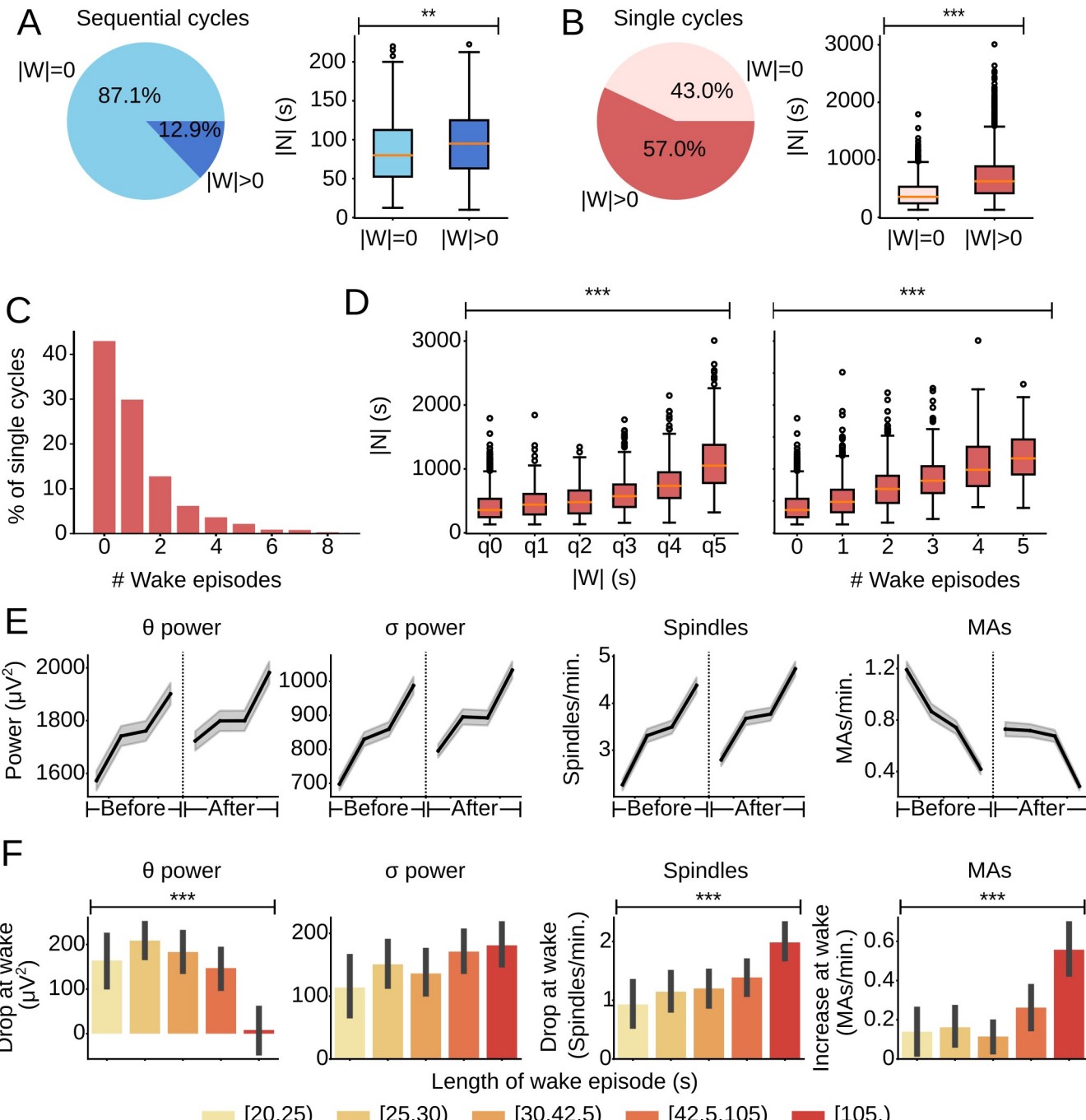

**Fig 5. Relationship between wake episodes and NREM sleep during an inter-REM interval.** (A) (Left) Pie chart indicating the percentage of sequential cycles without (|W| = 0) and with wake episodes (|W| > 0). (Right) Box plot comparing |N| for sequential cycles with and without wake episodes (t-test, t = -3.22, p = 0.0012, $n_{|W|>0}$ = 122, $n_{|W| = 0}$ = 825 cycles). (B) (Left) Pie chart indicating the percentage of single cycles without and with wake episodes. (Right) Box plot comparing |N| for single cycles with and without wake episodes (Welch's t-test, t = -28.64, p = 3.97e-163, $n_{|W|>0}$ = 2259, $n_{|W| = 0}$ = 1702). (C) Bar plot showing the distribution of the number of wake episodes during the inter-REM interval of single cycles. Note that 99.63% of single cycles had 8 or fewer wake episodes. (D) (Left) Box plot comparing total NREM duration, |N|, for single cycles with increasing values of total wake duration, |W| (Welch's ANOVA, F (5,1233.21) = 301.99, p = 3.78e-211). The x-tick q0 corresponds to cycles without wake. The remaining cycles with |W| > 0 were subdivided into quintiles, labeled q1—q5, based on the distribution of |W| for single cycles. (Right) Box plot comparing |N| for single cycles based on the number of wake episodes occurring during the inter-REM interval (Welch's ANOVA, F(5,475.26) = 246.53, p = 1.65e-129). Note that 97.61% of single cycles contained 5 or fewer wake episodes. (E) Progression of θ power, σ power, spindle rate, and MA rate during NREM sleep before and after a wake episode. Only sequences with at least 1 minute of NREM sleep both before and after wake during the inter-REM interval of single cycles were included. The duration of NREM episodes was normalized. 'Before' refers to all NREM sleep in between either the previous REM or wake episode and the current wake episode. 'After' refers to all NREM

sleep in between the current wake episode and either the next wake or REM episode. Shadings, 99% CI. (F) Bar plot showing average drop (or increase) in $\theta$ power, $\sigma$ power, spindle rate, and MA rate over wake episodes with different durations. All wake episodes for single cycles were divided into five quintiles based on the distribution of their durations. A drop or increase in each variable was calculated by subtracting the average value for 1 min of NREM after wake from the average value for 1 min of NREM before wake ($\theta$: Welch's ANOVA, $F(4,899.90) = 10.48$, $p = 2.65\text{e-}08$; $\sigma$: ANOVA, $F(4,1901) = 1.99$, $p = 0.093$; Spindles: ANOVA, $F(4,1901) = 6.22$, $p = 5.70\text{e-}05$; MAs: Welch's ANOVA, $F(4,895.63) = 9.13$, $p = 3.08\text{e-}07$). Error bars, 95% CI from 1,000 bootstrap iterations.

-0.22, $R^2 = 0.010$, $p = 0.0020$) than for single cycles (slope = -0.09, $R^2 = 0.0089$, $p = 5.07\text{e-}09$) (**Fig 6A**). As single cycles are interrupted by longer inter-REM intervals, the intervening periods of NREM sleep and wakefulness may themselves influence $REM_{post}$, thereby weakening the correlation between the two variables. Furthermore, the negative correlation supports the notion of a short-term hourglass process: As more REM propensity is discharged during a longer REM episode, the subsequent episode tends to be shorter.

Previous studies found that the duration of a REM episode also depends on the amount of NREM sleep preceding REM sleep. But this correlation is typically only weak [19,21,43,44] or was reported to be non-existent [20]. In our dataset, we found no significant correlation between the preceding inter-REM interval duration of single cycles (**S8A Fig**; slope = 0.0012, $R^2 = 4.92\text{e-}04$, $p = 0.17$) and $REM_{post}$, or between the amount of NREM, $|N|$, and $REM_{post}$ (**Fig 6B**; slope = 0.0041, $R^2 = 9.83\text{e-}04$, $p = 0.052$).

The CDF of the conditional GMM describes the probability of entering REM sleep within $|N|$ (s) of NREM sleep since the last REM sleep episode, and it can therefore be interpreted as a measure of the ultradian propensity for REM sleep throughout a single sleep cycle. For a given value of $REM_{pre}$, there is considerable variation in the values of $|N|$ during single cycles and we tested whether the resulting differences in the CDF values at REM onset may influence the subsequent REM sleep duration. We indeed found that $REM_{post}$ and the value of the CDF at REM onset were positively correlated (**Fig 6C**; slope = 22.82, $R^2 = 0.013$, $p = 1.01\text{e-}12$), suggesting that a higher propensity at REM onset leads to longer REM sleep episodes. This correlation was particularly pronounced for cycles with $REM_{pre} \geq 60$ s. The finding that the REM duration is more closely correlated with the CDF than with the preceding NREM duration (Williams' correlation test, $t = 17.59$, $p = 0.0$) suggests that it is not the amount of NREM sleep, but rather the propensity for REM sleep accumulated throughout the sleep cycle that influences the REM duration.

Next, to test whether the presence of wake periods affects the correlation between the CDF and REM duration, we calculated the correlation between the CDF and $REM_{post}$ separately for sleep cycles with and without wake episodes (**S8D Fig**). Importantly, the dependence of $REM_{post}$ on the CDF value was not affected by the presence of wake episodes. This finding suggests that the impact of the REM propensity, as quantified by the CDF, on $REM_{post}$ is not influenced by the presence of wake episodes.

Finally, we analyzed whether the CDF at REM onset also affects the EEG power during REM sleep. Changes in the EEG power were particularly pronounced for the prefrontal EEG (**Fig 6D**; see **S8B Fig** for parietal EEG). We found that if the animal transitioned to REM sleep at a low CDF value, the power of the prefrontal EEG in the $\delta$, $\theta$, and $\sigma$ range was higher than when REM sleep was entered at a high CDF value (**Fig 6D**). To test whether the negative correlation between the CDF and EEG power may be explained by differences in the REM sleep duration, we calculated linear approximations of the spectral densities for each CDF range based on the distribution of REM durations within this range (**S8C Fig and Methods**). For high values of the CDF (CDF $\geq 0.4$), the linear approximation closely matched the observed densities. However, for low CDF values (CDF $< 0.4$) the approximation considerably differed

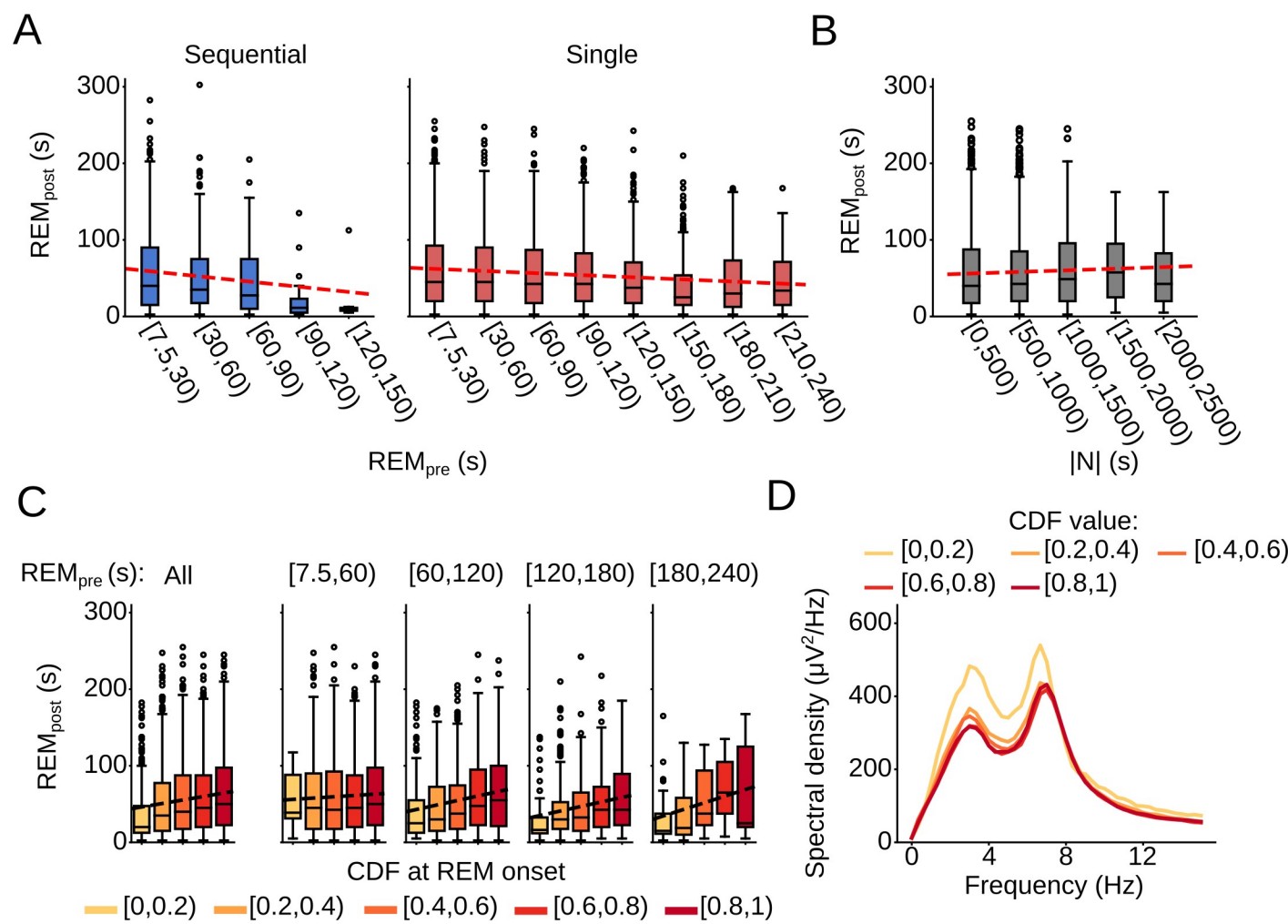

**Fig 6. Effects of sleep history on REM episode duration.** (A) (Left) Box plot comparing the duration of REM sleep ($REM_{post}$) following sequential cycles for different values of $REM_{pre}$. Red line, linear regression (slope = -0.22, $R^2$ = 0.010, p = 0.0020). (Right) Box plot comparing $REM_{post}$ following single cycles for different values of $REM_{pre}$. Red line, linear regression (slope = -0.091, $R^2$ = 0.0089, p = 5.07e-09). (B) Box plot comparing $REM_{post}$ following single cycles with different values of $|N|$. Red line, linear regression (slope = 0.0041, $R^2$ = 9.83e-04, p = 0.052). (C) Box plots comparing $REM_{post}$ dependent on the CDF value at REM onset for single cycles. The left plot includes all $REM_{pre}$ values; the remaining plots show the correlation for increasing ranges of $REM_{pre}$. Dashed lines, linear regression (All: slope = 22.82, $R^2$ = 0.013, p = 1.01e-12; [7.5,60): slope = 8.58, $R^2$ = 0.0013, p = 0.090; [60,120): slope = 29.42, $R^2$ = 0.024, p = 1.57e-07; [120,180): slope = 29.64, $R^2$ = 0.039, p = 3.12e-06; [180,240): slope = 43.28, $R^2$ = 0.077, p = 0.0060). (D) Spectral density of prefrontal EEG during REM episodes following single cycles as a function of the CDF at REM onset (Welch's ANOVA, δ: $F_{(4,1078.85)}$ = 16.18, p = 7.04e-13; θ: $F_{(4,1084.95)}$ = 6.06, p = 8.0e-05; σ: $F_{(4,1089.74)}$ = 9.46, p = 1.61e-07).

from the observed densities, suggesting that, at least in part, the magnitude of the prefrontal EEG power during REM sleep reflects REM sleep propensity.

## Changes in REM sleep regulation between light and dark phase

Next, we applied our model to test for differences in the regulation of REM sleep between the light and dark phase. As expected [45,46], mice spent less time in sleep during the dark phase (**Fig 7A**). In addition, the ratio of REM sleep to the total amount of sleep was reduced (**Fig 7B**). To further understand why the relative amount of REM sleep is reduced during the dark phase, we fit the conditional GMM to the recordings from the dark phase (**Figs 7D and S9A**). As was the case for the light phase, the Lilliefors-corrected KS test did not provide enough evidence to reject the null hypothesis that $|N|$ conditional on $REM_{pre}$ follows a lognormal Gaussian mixture distribution (**S5 and S6 Tables**).

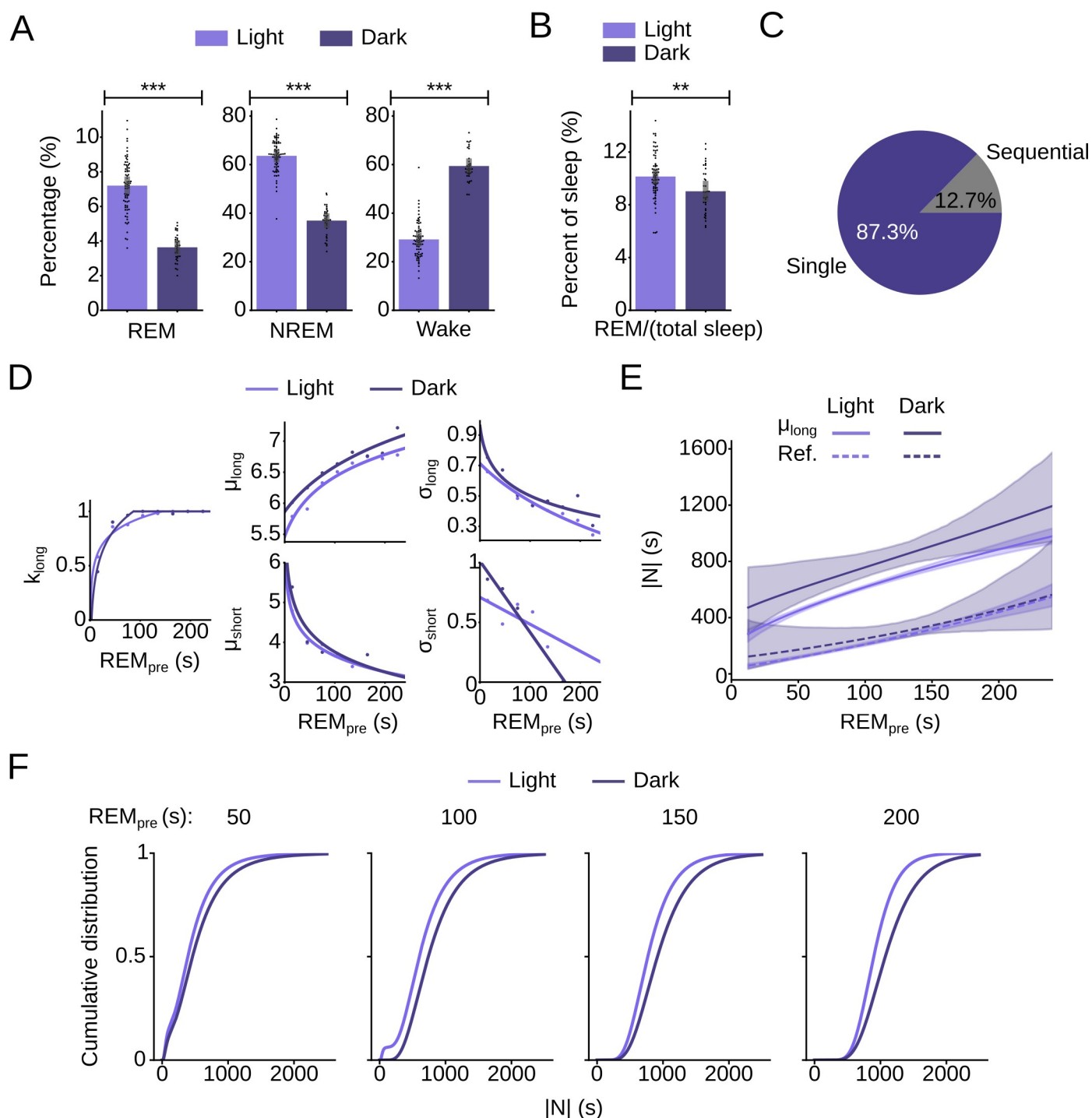

**Fig 7. Changes in REM sleep regulation between light and dark phase.** (A) Bar plots comparing the percentage of REM, NREM sleep, and wake during the light and dark phase (REM: Welch's t-test, t = 16.14, p = 3.37e-30; NREM: t-test, t = 20.26, p = 4.53e-38; Wake: t-test, t = -20.79, p = 5.19e-39; $n_{light}$ = 72, $n_{dark}$ = 35 mice). Error bars, 95% CIs from 1,000 bootstrap iterations. (B) Bar plot comparing the ratio REM/(REM+NREM) for the light and dark phase (t-test, t = 3.17, p = 0.0019, $n_{light}$ = 72, $n_{dark}$ = 35 mice). Error bars, 95% CIs from 1,000 bootstrap iterations. (C) Pie chart showing the percentage of sequential and single cycles during the dark phase. (D) Comparison of GMM parameters for the light and dark phase (Welch's t-test with Bootstrap, $k_{long}$: t = -77.26, p = 0.0; $\mu_{long}$: t = -372.77, p = 0.0; $\sigma_{long}$: t = -57.93, p = 0.0; **Methods**). (E) Comparison of $\mu_{long}$ (solid lines), and threshold (Ref., dashed lines) separating the refractory from the permissive period for the light and dark phase. Shadings, 95% CIs obtained from 10,000 bootstrap iterations. (F) Comparison of the CDFs of the GMMs for the light and dark phase shown for 4 different values of $REM_{pre}$.

Based on our model, we identified two major factors underlying the reduced REM/total sleep ratio during the dark phase. First, during the dark phase, the average of $k_{long}$, the weighting factor for the long Gaussian, was significantly increased (**Fig 7D and Methods**), reflecting the increased percentage of single cycles and reduction in sequential cycles (**Fig 7C**) compared to the light phase (**Fig 3B**). As single cycles contain less REM sleep relative to NREM sleep, their reduced frequency during the dark phase, mirrored in the increase of $k_{long}$, contributes to the decrease in REM sleep. Second, the mean of the long Gaussian distribution, $\mu_{long}$, was larger in the dark phase, indicating a delayed increase in the likelihood of NREM to REM sleep transitions, a second factor contributing to the increase in NREM relative to REM sleep (**Fig 7D and 7E**). Interestingly, the duration of the refractory period did not significantly differ between the light and dark phase (**Fig 7E**) because $\sigma_{long}$ was also increased during the dark phase (**Fig 7D**), counteracting the effects of $\mu_{long}$ on the duration of the refractory period. Consequently, for large values of $REM_{pre}$ ($\geq 150$ s), the CDFs for both the light and dark phase stayed close to zero for approximately the same amount of time but then increased at different rates (**Fig 7F**). Hence, our model suggests that the probability of sequential cycles is decreased during the dark phase and that the rate at which REM propensity increases throughout the sleep cycle is reduced during the dark phase.

## Discussion

In this study, we characterized the relationship between $REM_{pre}$ and subsequent NREM sleep using a conditional GMM. The distribution of NREM sleep during sleep cycles is bimodal, suggesting that two different types of inter-REM intervals exist (**Figs 1 and 2**). Using our model, we separated short from long inter-REM intervals and found that NREM sleep during short intervals displays reduced δ and increased θ power (**Fig 3**). Longer inter-REM intervals begin with a refractory period, during which transitions to REM sleep are highly unlikely (**Fig 4**). The refractory period proportionally increases in duration with $REM_{pre}$ and is characterized by a low θ power, σ power, and spindle rate as well as an increased frequency of MAs. The total duration of NREM sleep also depends on the number and total duration of wake periods during the inter-REM interval (**Fig 5**). $REM_{pre}$ is negatively correlated with the subsequent REM sleep duration. In addition, the CDF of the conditional GMM is positively correlated with the next REM duration, suggesting that a higher propensity for REM sleep results in longer REM episodes (**Fig 6**). The build-up of the REM propensity is delayed during the dark phase (**Fig 7**). Altogether, our analysis suggests that three major factors shape the ultradian regulation of REM sleep: The presence of two distinct types of sleep cycles, a refractory period suppressing transitions to REM sleep, and a propensity for REM sleep that influences the next episode duration.

### Sequential sleep cycles

Sequential sleep cycles have been observed in rats [32,33,35], cats [21], monkeys [37] and humans [36,47]. The presence of both short and long cycles results in a bimodal distribution of inter-REM intervals in these species. The minimum between the two modes for short and long inter-REM intervals served as a threshold to separate sequential from single cycles. Comparative analysis showed that this threshold and the average duration of inter-REM intervals increase with brain size [1,34]. In rats and, as shown here, in mice the durations of REM sleep episodes at the beginning of sequential cycles are on average shorter than those at the start of single cycles, and the frequency of sequential cycles is reduced during the dark period [33]. Thus, sequential cycles in both species share a lot of statistical similarities suggesting a common physiological mechanism in both species. Infusion of a serotonin receptor agonist into

the laterodorsal tegmentum reduced the frequency of REM sleep sequences, while an antagonist increased their occurrence in rats [48], suggesting a role of serotonin in their regulation. We speculate that an increased release of serotonin may contribute to the suppression of sequential sleep cycles during the dark phase [49,50].

Similar to our findings in mice, the EEG δ and σ power in rats is also reduced during sequential cycles [51]. A previous study in cats described NREM sleep during sequential cycles as light slow wave sleep as it is also characterized by low δ power [21]. NREM sleep during sequential cycles is thus reminiscent of the so-called intermediate stage preceding a transition to REM sleep, which shares both features of NREM and REM sleep [41,52]. For future studies, it would be interesting to perform simultaneous local field potential recordings from multiple sites to test whether both NREM and REM sleep states may locally co-exist in different brain areas during sequential cycles as observed for the intermediate stage [41,52].

Current dynamical systems models generate the ultradian alternation between NREM and REM sleep either through feedback from the arousal system to the circadian or homeostatic drive for sleep [53] or by implementing mutually inhibitory interactions between REM-on and REM-off neurons [54,55]. In the latter model type, a homeostatic hourglass process dictates the timing of REM sleep, an assumption supported by the positive correlation between $REM_{pre}$ and the following inter-REM interval. Testing under which assumptions these models can reproduce the lognormal distribution of NREM sleep may further constrain the time course at which REM propensity accumulates. For future studies, it would be interesting to investigate whether these dynamical models can also explain the generation of sequential cycles, possibly by introducing physiologically motivated noise terms mimicking fluctuations in firing rates or neurotransmitter concentrations [54,56].

## REM sleep is followed by a refractory period

As the conditional GMM allowed us to separate single from sequential sleep cycles, we could disentangle the refractory period, which only exists for single cycles, from sequential cycles and define it for the whole range of $REM_{pre}$. Since the refractory period proportionally increases with the duration of the preceding REM sleep episode (by about $2 \cdot REM_{pre}$), it may mechanistically explain the positive correlation between $REM_{pre}$ and succeeding NREM sleep [29–31]. With its low σ power, spindle rate, and high rate of MAs, the refractory period likely constitutes a fragile state of NREM sleep in mice. A study in cats similarly found that REM sleep is followed by a stage of light sleep [21]. In humans, the sleep phase N1, which is characterized by the absence of spindles, is most likely to occur during the sleep cycle after REM sleep [57] (published in preprint). Hence, the refractory period may be classified as a substage of NREM sleep in mice that resembles stage N1 in humans.

At present, the refractory period in our study is statistically defined as an interval during which REM sleep is unlikely to occur. The physiological mechanisms underlying this period, however, are unknown. To further characterize this substage of NREM sleep, it would be interesting to test whether activation of known REM sleep-promoting neurons specifically during the refractory period is indeed ineffective in inducing REM sleep. Electrophysiological recordings in the ventrolateral periaqueductal gray (vlPAG) indicated a role of GABAergic REM-off neurons in this area in the ultradian regulation of REM sleep. The activity of vlPAG REM-off neurons gradually decreases during the inter-REM interval, and abruptly rises at the end of REM sleep in a duration-dependent manner: The longer the REM episode lasts, the more these neurons become subsequently activated [13]; an effect which may mediate the dependence of the refractory period on the preceding REM duration [29–31]. Interestingly, vlPAG GABAergic neurons express orexin/hypocretin receptors [58] and may thus be excited by the

wake-active orexin/hypocretin neurons known to project to the vlPAG [17]. Such an excitatory drive may underlie the delayed increase in REM propensity during the dark phase suggested by our model: A higher baseline activity of REM-off neurons during the dark phase, when orexin/hypocretin levels are high [59], may delay the time till the activity of these neurons is low enough to allow for a transition to REM sleep.

Our analysis suggests a close association of θ, σ power, and sleep spindles with the probability of NREM to REM transitions. The spindle rate is reduced during the refractory period, when transitions to REM sleep are highly unlikely, while the increased rate of spindles during the permissive period may facilitate REM sleep. In support of this view, a recent study showed that optogenetically triggering spindles through stimulation of the thalamic reticular nucleus enhances the chance of transitions to REM sleep [42]. For future studies, it is therefore important to understand how thalamocortical circuits generating spindles interact with hypothalamic or brainstem circuits controlling REM sleep.

### Interaction between REM propensity and REM episode duration

As the REM sleep duration is negatively correlated with the occurrence of sequential cycles and positively correlated with the duration of the refractory period, it plays a crucial role in temporally shaping the sleep cycle. Previous studies showed that during the homeostatic rebound following REM sleep deprivation, REM sleep episodes are elongated [28,60,61], suggesting that an increased propensity for REM sleep is associated with longer episodes. We similarly found that during spontaneous sleep a higher REM propensity, as reflected in the CDF of the conditional GMM, results in longer REM episodes. Our finding suggests that the REM propensity builds up fast enough during the sleep cycle to influence the duration of REM sleep episodes. Consistent with this, short-term REM sleep deprivation (as short as 20 min) induces a detectable rebound in REM sleep, further supporting the notion that REM sleep propensity accumulates at a time scale relevant to modulate the ultradian timing of REM sleep [60–62].

In addition to the episode duration, the CDF was also correlated with the EEG power during REM sleep. Interestingly, sleep recordings in humans also showed a reduction in the REM EEG power within a similar range (α power), after enhancing the REM propensity through REM sleep deprivation [63–65]. A recent study in rats found that the θ power declines throughout REM sleep and this decline is positively correlated with the amount of NREM sleep during the preceding inter-REM interval, also supporting a close association between REM propensity and changes in the EEG power [66].

For future studies, it would be interesting to test how the propensity for REM sleep reflected in the CDF of the conditional GMM relates to the homeostatic long-term process mediating the rebound in REM sleep following long-term total sleep or REM sleep deprivation. The long-term process regulates the daily quota of REM sleep and is thought to accumulate in the absence of REM sleep during both NREM sleep and wakefulness [29,67,68], and has been proposed to be separate from the short-term process regulating the ultradian timing of REM sleep [28]. In vivo recordings of vlPAG GABAergic neurons showed that the firing rates of these REM-off neurons during inter-REM are modulated by the preceding REM sleep duration, suggesting that they mirror the short-term REM propensity [13]. Recording the same neurons throughout long-term REM sleep deprivation and recovery sleep may provide important insights into the extent to which the short- and long-term processes are disjunct or overlap at the neuronal level.

### Role of Wakefulness

In addition to the preceding REM sleep, the total amount of NREM sleep during the sleep cycle is also modulated by wake periods. Similar to REM sleep, although to a lesser degree,

wake episodes lead to a decrease in the θ power, σ power, and spindle rate and an increase in the MA rate, quantities which are indicative of the likelihood of NREM to REM sleep transitions. Wake episodes may result in a reduction in the probability of REM transitions and thus lead to more NREM sleep, which in turn could explain the positive correlation between the total duration or number of wake episodes and the total NREM duration. Previous studies also suggest that an increased δ power induced by long intervals of wakefulness enforced by sleep deprivation suppresses the initiation of REM sleep [29,69]. On the ultradian time scale, however, we observed no systematic relation between the time course of the δ power and the likelihood of NREM to REM transitions.

Our finding that the presence or absence of wakefulness during the sleep cycle does not affect the correlation between the CDF value and subsequent REM duration (**S8D Fig**), indicates that, at least on the ultradian timescale, REM propensity and its impact on the REM episode duration is more closely associated with the time spent in NREM sleep than with the combined time in NREM sleep and wake. Thus, while wake periods reduce the following θ power, σ power, and rate of sleep spindles and may thereby lower the subsequent opportunity for NREM to REM transitions [32], they do not appear to interfere with the build-up of the REM propensity reflected in the CDF, reinforcing the notion that the ultradian REM propensity primarily accumulates during NREM sleep [20]. Altogether, our model-based approach provides a flexible framework to study the key factors underlying the ultradian timing of REM sleep and will inform future experimental studies to understand the mechanisms regulating the REM sleep duration, refractory period, and induction of REM sleep sequences.

## Methods

### Experimental setup

**Ethics statement.**   All experimental procedures were approved by the Institutional Animal Care and Use Committee (IACUC) at the University of Pennsylvania and conducted in accordance with the National Institutes of Health Office of Laboratory Animal Welfare Policy.

**Animals.**   Experiments were performed in male or female C57BL/6J mice (Jackson Laboratory stock no. 000664). Animals were housed on a 12-h dark/12-h light cycle (lights on between 7 am and 7 pm) and were aged 6–12 weeks at the time of surgery. All mice were group-housed with ad libitum access to food and water.

**Surgery.**   All surgeries were performed following the IACUC guidelines for rodent survival surgery. Prior to surgery, mice were given meloxicam subcutaneously (5 mg/kg). Mice were anesthetized using isoflurane (1–4%) and positioned in a stereotaxic frame. Animals were placed on a heating pad to maintain the body temperature throughout the procedure. Following asepsis, the skin was incised to gain access to the skull. For EEG recordings, stainless steel wires were attached to two screws, one on top of the parietal and one on top of the prefrontal cortex. The reference screw was inserted on top of the cerebellum. For EMG recordings, two stainless steel wires were inserted into the neck muscles. All electrodes, screws, and the mini-connector holding the EEG, EMG wires were secured to the skull using dental cement. After the injection and implantation were finished, bupivacaine (2 mg/kg) was administered at the incision site.

**Sleep recordings.**   Sleep recordings were performed in the animal's home cage or in a cage to which the mouse was habituated for 3 days, which was placed within a sound-attenuating box. EEG and EMG signals were recorded using an RHD2000 amplifier (intan, sampling rate 1 kHz). EEG and EMG signals were referenced to the common ground screw on top of the cerebellum. During the recordings, EEG and EMG electrodes were connected to flexible recording cables using a small connector. To determine the brain state of the animal, we first

computed the EEG and EMG spectrogram with consecutive fast Fourier transforms (FFTs) calculated for sliding, half-overlapping 5 s windows, resulting in a 2.5 s time resolution of the hypnogram. Next, we computed the time-dependent delta, theta, sigma, and high gamma power by integrating frequencies in the range 0.5 to 4 Hz, 5 to 12 Hz, 12 to 20 Hz, and 100 to 150 Hz, respectively. We also calculated the ratio of the theta and delta power (theta/delta) and the EMG power in the range 50 to 500 Hz. For each power band, we used its temporal mean to separate it into a low and high part (except for the EMG and theta/delta ratio, where we used the mean plus one standard deviation as threshold). REM sleep was defined by high theta/delta ratio, low EMG, and low delta power. A state was set as NREM sleep, if delta power was high, the theta/delta ratio was low and EMG power was low. In addition, states with low EMG power, low delta, but high sigma power were scored as NREM sleep. Wake encompassed states with low delta power and high EMG power and each state with high gamma power (if not otherwise classified as REMs). Of the bins classified as wake periods, those forming sequences of 20 s or less were classified as microarousals. Finally, we manually rescored the automatic classification to correct for errors using a graphical user interface, visualizing the raw EEG, EMG signals, spectrograms, and hypnogram. The software for automatic brain state classification and manual scoring was programmed in Python. The light-phase data contained 5098 sleep cycles from 125 recordings of 72 mice. The dark-phase data contained 1242 sleep cycles from 55 recordings of 35 mice.

## Data analysis

**Gaussian mixture model parameters.** Following the definition in [40], a sleep cycle comprises a REM sleep episode and the following inter-REM interval. All sleep cycles with REM sleep episode duration $REM_{pre} < 240 \, s$ were divided into 8 groups based on $REM_{pre}$. We chose 30 s non-overlapping bins ([0,30), [30,60), . . ., [210,240)) to ensure that each group contained enough cycles to reliably estimate the model parameters while being able to capture the change in these parameters conditional on $REM_{pre}$. For each group, we used the Expectation-Maximization algorithm on $\ln(|N|)$ to find the maximum likelihood estimates for $k_{long}$, $k_{short}$, $\mu_{long}$, $\mu_{short}$, $\sigma_{long}$, and $\sigma_{short}$. In the case of $REM_{pre} \geq 150 \, s$, the distribution of $\ln(|N|)$ was unimodal; consequently, $k_{long} = 1$ and we only estimated $\mu_{long}$ and $\sigma_{long}$. For consistency, we assumed that the apparent unimodality of the distribution for $REM_{pre} < 30 \, s$ results from the blending of the distributions for short and long cycles. Computations were performed using the python package scikit-learn [70].

**Lilliefors-corrected KS test.** The standard Kolmogorov-Smirnov (KS) test compares data to a pre-defined distribution by comparing the CDF to the empirical cumulative distribution function (ECDF). However, it is invalid when the parameters for the CDF are estimated using the data [71]. Therefore, to perform a valid goodness-of-fit test with the estimated parameters, we applied the Lilliefors-corrected KS test as follows:

1. For each consecutive 30 s bin of $REM_{pre}$, using the estimated parameters, we draw a random sample with the same size as the original data and calculate the KS-statistic.

2. We repeat (1) 10,000 times and form a distribution of KS-statistics, $KS_{sim}$.

3. We calculate the KS-statistic using the observed data ($KS_{obs}$) and find how extreme $KS_{obs}$ is when compared to the distribution of the simulated KS-statistics.

4. If $KS_{obs}$ is too large ($KS_{obs} > 95\%$ of $KS_{sim}$), we reject the null hypothesis that our data comes from the specified distribution.

Performing the Lilliefors-corrected KS test, we found that there is not enough evidence to reject the hypothesis that our data is drawn from a Gaussian mixture distribution (**S2 Table**). This was the case for both the light and dark phase and held true for different thresholds to score MAs (**S3 Table**).

**Conditional Gaussian mixture model.** After estimating all 6 parameters of the GMM for each 30 s bin of $REM_{pre}$, we used either a linear ($y = ax+b$) or logarithmic ($y = a \cdot ln(x+b)+c$; $b \geq 0$) function to model the relationship between each parameter and $REM_{pre}$ (**Figs 2D and S1A**). $k_{short}$ was implicitly defined as $k_{short} = 1-k_{long}$. Because $k_{long} = 1$ for $REM_{pre} > 150$ s, we only used the first 6 values of $k_{long}$ for fitting the function. We calculated the residual sum of squares (RSS) for both the linear and logarithmic fits and chose the function for which the RSS was lower. The fitting was performed using the Trust Region Reflective algorithm implemented in SciPy [72]. The logarithmic fit was generally better for all parameters except for $\sigma_{short}$. In total, the model comprises 14 parameters. The complete probability model can be expressed as follows:

$$P(ln(|N|)|REM_{pre} = x) = k_{long} \cdot f(x; \mu_{long}(x), \sigma^2_{long}(x)) + k_{short}(x) \cdot f(x; \mu_{short}(x), \sigma^2_{short}(x))$$

where $f$ is the probability density function of a Gaussian distribution, and

$$k_{long}(x) = min(a_{k,long} \cdot ln(x + b_{k,long}) + c_{k,long}, 1)$$

$$k_{short}(x) = 1 - k_{long}(x)$$

$$\mu_{long}(x) = a_{\mu,long} \cdot ln(x + b_{\mu,long}) + c_{\mu,long}$$

$$\mu_{short}(x) = a_{\mu,short} \cdot ln(x + b_{\mu,short}) + c_{\mu,short}$$

$$\sigma_{long}(x) = a_{\sigma,long} \cdot ln(x + b_{\sigma,long}) + c_{\sigma,long}$$

$$\sigma_{short}(x) = a_{\sigma,short} \cdot x + b_{\sigma,short}$$

Note that $k_{long}(x)$ is defined as the minimum of the result of the log function and 1, as $k_{long}(x)$ is a probability and cannot exceed 1.

For the dark phase, the data for $150 \leq REM_{pre} < 180$ contained only one single $ln(|N|)$ value falling into the short Gaussian distribution, resulting in a value for $k_{long}$ below 1. To avoid that a single data point changes the entire relationship between $REM_{pre}$ and $k_{long}$, we fit the logarithmic and linear functions on the first 4 values of $k_{long}$ (**Figs 7D and S9A**).

Of note, the complete conditional GMM for the light phase is defined only for sleep cycles with $REM_{pre} \geq 7.5$ s, because $REM_{pre} = 7.5$ s is the lowest value of $REM_{pre}$ for which the intersection point of the short and long Gaussian distributions, $x_{intersect}$, satisfies $\mu_{short} < x_{intersect} < \mu_{long}$ (**S1B Fig**). Similarly, the complete model for the dark phase is only defined for sleep cycles with $REM_{pre} \geq 12.5$ s (**S9B Fig**).

**Model simulation.** To assess the goodness-of-fit of the model to the data, we performed a simulation as follows:

1. For each $REM_{pre}$ in the data set with $7.5$ s $\leq REM_{pre} < 240$ s:

   a. Calculate $k_{long}$, $k_{short}$, $\mu_{long}$, $\mu_{short}$, $\sigma_{long}$, $\sigma_{short}$ using the conditional GMM.

   b. Based on these parameters, define the Gaussian mixture distribution for the given $REM_{pre}$ and sample one data point ($ln(|N|)$) from this distribution.

2. Perform step (1) 10,000 times.

3. Compare the simulated distribution of $ln(|N|)$ to the actual distribution of $ln(|N|)$ for sleep cycles with $REM_{pre}$ in the range $7.5\ s \leq REM_{pre} < 240\ s$.

We found that the simulated distribution of $ln(|N|)$ closely overlaps with the distribution of the actual data (**S1C Fig**).

**Single and sequential cycles.**   For each sleep cycle with $REM_{pre}$ in the range $7.5\ s \leq REM_{pre} < 240\ s$, we label the cycle as either single or sequential using the following procedure:

1. Using the conditional GMM, determine for $REM_{pre}$ the probability distributions for the short and long Gaussian distributions.

2. Find the intersection point, $x_{intersect}$, along the x-axis ($ln(|N|)$ axis) of the PDFs of the two Gaussian distributions.

3. If $ln(|N|) < x_{intersect}$, the cycle is sequential, otherwise it is labeled as single.

We use the criterion in step (3) because if $ln(|N|) < x_{intersect}$, the PDF of the short Gaussian evaluated at $ln(|N|)$ is larger than the PDF of the long Gaussian evaluated at $ln(|N|)$ and vice versa. i.e.

$$ln(|N|) < x_{intersect} \Longrightarrow k_{short} \cdot f_{short}(ln(|N|)) > k_{long} \cdot f_{long}(ln(|N|))$$

$$ln(|N|) < x_{intersect} \Longrightarrow k_{short} \cdot f_{short}(ln(|N|)) < k_{long} \cdot f_{long}(ln(|N|))$$

where $f_{short}$ and $f_{long}$ correspond to the PDFs of the short and long Gaussian distributions.

**Refractory and permissive period.**   For each value of $REM_{pre}$ in the range $7.5\ s \leq REM_{pre} < 240\ s$, we determine the threshold separating the refractory from the permissive period as such:

1. Using $REM_{pre}$ and the conditional GMM, calculate $\mu_{long}$ and $\sigma_{long}$.

2. Using these parameters, define the CDF of the long Gaussian distribution, $F_{long}(x)$.

3. The $ln(|N|)$ value for which $F_{long}(ln(|N|)) = 0.01$ is set as the threshold.

**Sleep spindle detection.**   For spindle detection, we first calculated the spectrogram for the prefrontal EEG. The spectrogram was computed for consecutive 600 ms windows with 500 ms overlap, resulting in a 100 ms temporal resolution. The spindle detection algorithm used two criteria to determine for each 100 ms time bin whether it was part of a spindle or not: The first criterion was that the height of the maximum peak in the sigma frequency range (10–16.67 Hz) exceeds a threshold, which corresponded to the 96th percentile of all maximum peaks in the sigma frequency range of the sleep recording. We determined the optimal percentile value by maximizing the performance of the algorithm on a manually annotated control data set. Second, the power value of the peak in the sigma range (10–16.67 Hz) had to be greater than half of the peak value in the range 0–10 Hz. The optimal value for this ratio (sigma peak ratio) was again determined on the control data set. Next, the algorithm merged spindle events that were temporally close to each other. First, spindle events in adjacent bins were considered as part of the same spindle. Second, we fused together sequences of spindle events that were interrupted by gaps of less than 300 ms. The optimal value for the gap was again determined on the control data set. Finally, we discarded spindles with duration $\leq$ 200 ms. Of all the potential spindles, we only considered those as spindles where for at least half of the time bins the peak

frequency lied in the range of 10–16.7 Hz. The parameters of the spindle detection algorithm (sigma percentile threshold, sigma peak ratio, and minimum fusing distance) were optimized using a manually annotated data set. The algorithm correctly identified 88.8% of spindles present in the annotated data set with a false positive rate of 6.9%.

**Linear approximation of spectral densities.** To test if differences in the EEG spectral density between single and sequential REM sleep episodes were due to differences in their respective durations (S3 Fig), we first discretized the distribution of $REM_{pre}$ for both types of cycles into eight 30 s bins and calculated for each bin the corresponding REM sleep duration-dependent spectral density, $P_{dur,i}$. Then, for single and sequential cycles, we found the proportion of cycles, $w_{single,i}$ and $w_{seq,i}$, with $REM_{pre}$ falling into bin $i$. Note,

$$\sum_i w_{single,i} = 1, \text{ and } \sum_i w_{seq,i} = 1 \text{ for } i \in \{1, \ldots, 8\}.$$

Using these proportions as weights, we calculated a weighted average of the duration dependent spectral densities to give us linear approximations, $\hat{P}_{seq}$ and $\hat{P}_{single}$ of the true spectral densities as follows:

$$\hat{P}_{single} = \sum_i w_{single,i} \cdot P_{dur,i} \text{ and } \hat{P}_{seq} = \sum_i w_{seq,i} \cdot P_{dur,i}$$

Similarly, we tested whether differences in the EEG during REM sleep were due to differences in the CDF at REM onset or the result of differences in $REM_{pre}$. For each of the five groups determined by the CDF value at REM onset, we found the proportion of REM periods falling into each bin and used these proportions as weights for the weighted averages (S8C Fig).

**Spectral density and power estimation.** The EEG and EMG signals were sampled at 1000 Hz. The hypnogram was binned in 2.5 s epochs. Spectral densities of the EEG were computed using Welch's method with a Hanning window for 3 seconds long, half overlapping intervals, resulting in a frequency resolution of 1/3 Hz. Frequency bands were defined as follows: δ: 0.5–4.5 Hz, θ: 5–9.5 Hz, σ: 10–15 Hz. To calculate the power for each frequency band, we approximated the corresponding area under the spectral density curve using a midpoint Riemann sum.

**Bootstrap comparison of light and dark phase GMM parameters.** We performed bootstrapping with 10,000 iterations on the entire data set for both the light and dark phase and estimated the GMM parameters using the same procedure as explained above. To compare differences between the two phases, we computed for $k_{long}$, $\mu_{long}$, and $\sigma_{long}$ the log-function relating each parameter to $REM_{pre}$ and then determined the average value of that function over the entire range of $REM_{pre}$ for which the conditional GMM is defined. Repeating this procedure for each bootstrap iteration resulted in two distributions for each parameter corresponding to the light and dark phase, respectively. Welch's t-test revealed significant differences between the light and dark phases ($k_{long}$: t = -77.26, p = 0.0; $\mu_{long}$: t = -372.77, p = 0.0, $\sigma_{long}$: t = -57.93, p = 0.0).

**Statistics.** Statistical tests were performed using the python modules scipy [72] and pingouin [73] and the R package cocor [74]. Linear regressions were performed with the python module statsmodels [75]. For comparisons of quantities between two groups, we used the Levene test to check the homoscedasticity of the data and performed either t-tests or Welch's t-tests. To compare quantities between multiple groups, the data sets were compared using either one-way or two-way ANOVA followed by multiple comparisons tests.

## Supporting information

**S1 Fig. Conditional GMM for the light phase and model validation.** (A) Comparison of logarithmic (solid lines) and linear (dashed lines) fits for the functions describing the relationship between $REM_{pre}$ and each GMM parameter. For $k_{long}$, we only fitted the functions to the first 6 $REM_{pre}$ bins (filled circles). For the remaining $REM_{pre}$ bins, $k_{long}$ was set to 1. (B) Estimated probability density functions (PDFs) of the Gaussian distributions for short and long cycles for each 2.5 s increment of $REM_{pre}$ in the range 2.5 s—12.5 s. (C) Histogram over $|N|$ on the normal (left) and natural log scale (right). The histogram for the actual data is compared with the prediction by the GMM (10,000 model simulations).
(PDF)

**S2 Fig. Mean and standard deviation of inter-REM and $|N|$ as a function of $REM_{pre}$.** (A) Mean and standard deviation of ln(inter-REM) as a function of $REM_{pre}$ (n = 5090). (B) Mean and standard deviation of ln($|N|$) as a function of $REM_{pre}$.
(PDF)

**S3 Fig. Prefrontal EEG during REM sleep for sequential and single cycles.** (A) Spectral density of prefrontal EEG during REM sleep for different REM sleep durations ($REM_{pre}$). (B) Spectral density of prefrontal EEG during REM sleep for sequential and single cycles. Solid lines represent the actual densities with shadings indicating the 99% CIs. The dashed lines represent the weighted averages of the duration-dependent densities in A. The weighted averages were calculated based on the proportion of $REM_{pre}$ values falling into each 30 s bin (**Methods**).
(PDF)

**S4 Fig. Comparison of different MA thresholds (1).** (A) Pie chart showing the percentage of REM, NREM, and Wake for different MA thresholds. All wake episodes with duration $\leq$ 30 s or $\leq$ 10 s were scored as MA; for the threshold of 0 s, no MAs were scored. (B) Spectral density of parietal (top) and prefrontal EEG (bottom) during NREM sleep for both sequential and single cycles using different MA thresholds. Horizontal lines indicate frequencies at which the spectral densities for sequential and single cycles are statistically different (Welch's t-test, *** p<0.001). Shadings, 99% CI. (C) Progression of θ power, σ power, and MA rate throughout the refractory and permissive period for different MA thresholds. The durations of both the refractory and permissive period were normalized to unit length and subdivided into quartiles of equal normalized duration. The average for all $REM_{pre}$ values is shown in black. Shadings, 99% CI.
(PDF)

**S5 Fig. EEG power, spindle rate, and MA rate throughout the refractory and permissive period.** (A) Spectral density of the parietal EEG for NREM sleep during the refractory and permissive period. Horizontal lines indicate frequencies at which the spectral densities for sequential and single cycles are statistically different; (Welch's t-test, *** p<0.001, $n_{refractory}$ = $n_{permissive}$ = 3934). (B) Progression of δ power throughout the refractory and permissive period for different ranges of $REM_{pre}$. The average for all $REM_{pre}$ values is shown in black. Shadings, 99% CI. (C) Progression of θ power, σ power, rate of spindles, and rate of MAs on normalized time scale throughout NREM sleep. Similar to **Fig 4F** but with the last 40 s of NREM sleep before the onset of the next REM episode excluded. (D) Progression of spindle rate and MA rate on the non-normalized time scale throughout the first 600 s of NREM sleep during the inter-REM interval for different values of $REM_{pre}$. The two vertical dashed lines indicate the lowest and highest threshold separating the refractory from the permissive period derived from the low and high bounds of the corresponding $REM_{pre}$ range. Shadings, 99% CI. (E)

Progression of θ power, σ power, spindle rate, and MA rate on the non-normalized time scale throughout the first 600 s of NREM sleep after a wake period. Different ranges of wake episode durations were selected to match the corresponding durations of $REM_{pre}$ in D. Shadings, 99% CI.
(PDF)

**S6 Fig. Comparison of different MA thresholds (2).** (A) Box plots comparing total NREM duration, |N|, for single cycles with increasing values of total wake duration, |W|, for different MA thresholds (*30 s*: Welch's ANOVA, $F_{(5,832.43)} = 320.67$, $p = 2.87e-191$; *10 s*: Welch's ANOVA, $F_{(5,1850.87)} = 359.09$, $p = 2.22e-269$; *No MA*: Welch's ANOVA, $F_{(4,1951,53)} = 1113.21$, $p = 0.0$). The x-tick q0 corresponds to cycles without wake. The remaining cycles with |W| > 0 were subdivided into quintiles, labeled q1—q5, based on the distribution of |W| for single cycles. For the threshold of 0 s, there was no cycle without wake. (B) Box plots comparing |N| for single cycles based on the number of wake episodes occurring during the inter-REM interval (*30 s*: Welch's ANOVA, $F_{(5,249.10)} = 281.68$, $p = 2.54e-100$; *10 s*: Welch's ANOVA, $F_{(8,511.64)} = 202.86$, $p = 2.55e-153$; *No MA*: Welch's ANOVA, $F_{(19,681.55)} = 448.18$, $p = 0.0$). (C) Progression of θ power, σ power, spindle rate, and MA rate during NREM sleep before and after a wake episode for different MA thresholds. Only sequences with at least 1 minute of NREM sleep both before and after wake during the inter-REM interval of single cycles were included. The duration of NREM episodes was normalized. 'Before' refers to all NREM sleep in between either the previous REM or wake episode and the current wake episode. 'After' refers to NREM sleep in between the current wake episode and either the next wake or REM episode. Shadings, 99% CI.
(PDF)

**S7 Fig. Relationship between wake episodes and NREM sleep.** (A) Scatter plots for |W| vs. |N| during single cycles for an increasing number (1–10) of wake episodes during the inter-REM interval. For each number of wake episodes, |W| and |N| are positively correlated. Red lines, linear regression fits. (B) Scatter plots for the number of wake episodes vs. |N| during single cycles for increasing ranges of |W|. In each case, the number of wake episodes and |N| was positively correlated. Red lines, linear regression fits.
(PDF)

**S8 Fig. Variables influencing REM episode duration and EEG.** (A) Box plot comparing the duration of REM sleep ($REM_{post}$) following single cycles for different values of inter-REM. Red line, linear regression (slope = 0.0012, $R^2$ = 4.92e-04, p = 0.17). (B) Spectral density of parietal EEG during REM episodes following single cycles for different values of the CDF at REM onset (δ: ANOVA, $F_{(4,3794)} = 1.27$, p = 0.27; θ: Welch's ANOVA, $F_{(4,1158.22)} = 16.18$, p = 6.65e-13; σ: Welch's ANOVA, $F_{(4,1161.57)} = 16.27$, p = 5.62e-13). (C) Comparison of the true and estimated spectral densities of REM periods for different ranges of the CDF values at REM onset. Solid lines represent the true spectral densities with shadings representing the 99% CIs. Dashed lines indicate estimated weighted averages (**Methods**). (D) Comparison of the relationship between the CDF (REM propensity) at REM onset and $REM_{post}$ for single cycles with wake (|W| > 0) and single cycles without wake (|W| = 0). The first row shows box plots comparing $REM_{post}$ of single cycles with |W| > 0 with different CDF values at REM onset. The first column shows the correlation for the full range of $REM_{pre}$; the remaining columns display the results for different bins of $REM_{pre}$. The second row shows the same comparisons for single cycles with |W| = 0. Dashed lines, linear regression ('s' indicates the slope of the regression line). The third row shows the linear regression results of CDF vs. $REM_{post}$ for single cycles with |W| > 0 and single cycles with |W| = 0. Shadings indicate 95% CIs obtained from 10,000

bootstrap iterations. Columns correspond to different bins of $REM_{pre}$.
(PDF)

**S9 Fig. Conditional GMM for the dark phase.** (A) Comparison of logarithmic and linear fits for the functions describing the relationship between $REM_{pre}$ and each GMM parameter for the dark phase. For $k_{long}$, the functions were only fitted to the first 4 $REM_{pre}$ splits (filled circles). (B) Estimated PDFs of the short and long Gaussian distributions for each 2.5 s increment of $REM_{pre}$ in the range 5 s—15 s for the dark phase.
(PDF)

**S1 Table. Inter-individual variability.** Mean and standard deviation across animals of key variables in the sleep pattern of mice during the light and dark phase. Note that the $R^2$ values for $REM_{pre}$ vs. |N|, |W|, or inter-REM differ from those in **Fig 1B**, as linear regression was performed for each animal individually before averaging, instead of computing $R^2$ values for the whole data distribution from all animals.
(PDF)

**S2 Table. GMM parameters for the light phase.** The columns indicate for each range of $REM_{pre}$ the number of sleep cycles, the weight of the long Gaussian distribution, mean and standard deviation of the short and long Gaussian distribution and the p-value obtained for the Lilliefors-corrected Kolmogorov-Smirnov (KS) test **(Methods).**
(PDF)

**S3 Table. P-values from Lilliefors-corrected KS-test for GMMs estimated for different MA thresholds.** The GMM was fitted on the light phase data set for varying thresholds used to score MAs. The columns show the p-values of the Lilliefors-corrected KS-test for the different MA thresholds.
(PDF)

**S4 Table. Coefficients of the conditional GMM for the light phase.** Each column shows the coefficients (a,b,c) for the logarithmic or linear functions describing each GMM parameter as a function of $REM_{pre}$ **(Methods).** For all parameters, we used a logarithmic fit except for $\sigma_{short}$.
(PDF)

**S5 Table. GMM parameters for the dark phase.** The columns indicate for each range of $REM_{pre}$ the number of sleep cycles, the weight of the long Gaussian distribution, mean and standard deviation of the short and long Gaussian distribution and the p-value obtained for the Lilliefors-corrected KS test for the dark phase **(Methods).**
(PDF)

**S6 Table. Coefficients of the conditional GMM for the dark phase.** Each column shows the coefficients (a,b,c) for the logarithmic or linear functions describing each GMM parameter as a function of $REM_{pre}$ for the dark phase (**Methods**). For all parameters, we used a logarithmic fit except for $\sigma_{short}$.
(PDF)

## Author Contributions

**Conceptualization:** Sung-Ho Park, Franz Weber.

**Data curation:** Sung-Ho Park, Justin Baik, Benjamin Kurland.

**Formal analysis:** Sung-Ho Park.

**Funding acquisition:** Franz Weber.

**Investigation:** Sung-Ho Park, Jiso Hong, Hanna Antila.

**Methodology:** Sung-Ho Park.

**Project administration:** Franz Weber.

**Software:** Sung-Ho Park, Justin Baik, Benjamin Kurland.

**Supervision:** Shinjae Chung, Franz Weber.

**Validation:** Benjamin Kurland.

**Visualization:** Sung-Ho Park.

**Writing – original draft:** Sung-Ho Park, Franz Weber.

**Writing – review & editing:** Sung-Ho Park, Franz Weber.

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
