## [Decision Letter · Decision Letter 0]

30 Apr 2021

Dear Dr Weber,

Thank you very much for submitting your manuscript "A probabilistic model for the ultradian timing of REM sleep in mice" for consideration at PLOS Computational Biology. As with all papers reviewed by the journal, your manuscript was reviewed by members of the editorial board and by several independent reviewers. The reviewers appreciated the attention to an important topic. Based on the reviews, we are likely to accept this manuscript for publication, providing that you modify the manuscript according to the review recommendations.

Sincerely,

Paul Franken

Guest Editor

PLOS Computational Biology

Wolfgang Einhäuser

Deputy Editor

PLOS Computational Biology

[LINK]

Reviewer's Responses to Questions

**Comments to the Authors:**

**Please note that the review of reviewer #2 is uploaded as attachment.**

Reviewer #1: This paper addresses an important unsolved problem in sleep research: the genesis of REM/NREM sleep cycles. The authors find that the dynamics in mice can be adequately explained in terms of a bifurcation between sequential REM bouts (i.e., essentially a continuation of the REM state) and a refractory period that separates distinct REM bouts, with the length of this period being dependent on the length of the preceding REM bout. The paper provides a powerful treatment of this issue. I applaud the authors for being so incredibly thorough with their analysis of the data. This approach allows a number of alternate interpretations to be directly tested. Overall I enjoyed reading the paper. Please see my suggestions below.

1) Regarding the sequential bouts, the authors muse on the idea that the interceding NREM bouts appear to have some features of REM and thus may be a sort of intermediate state. Have the authors considered this pattern in the context of local sleep?

2) The genesis of REM/NREM sleep cycles has been considered previously using dynamical systems models. These models have helped to propose and test potential theoretical mechanisms. Specifically, see:

Diniz Behn CG, Ananthasubramaniam A, Booth V. Contrasting existence and robustness of rem/non-rem cycling in physiologically based models of rem sleep regulatory networks. SIAM Journal on Applied Dynamical Systems. 2013;12(1):279-314. 

Phillips AJ, Robinson PA, Klerman EB. Arousal state feedback as a potential physiological generator of the ultradian REM/NREM sleep cycle. Journal of theoretical biology. 2013 Feb 21;319:75-87. 

It would be valuable to discuss the present findings in the context of these candidate models. I can imagine that the sequential states could be readily explained within a dynamical model by introducing stochasticity, as this could cause temporary transit between REM and NREM sleep, even before a REM sleep bout has really concluded. The positive association between REMpre and N is clearly in line with models that assume a homeostatic balance of the stages (and probably not in line with models that assume a clock-like mechanism). However, it is less clear to me whether the findings for W are consistent with existing dynamical models, or whether models would need to be amended to explain these results.

3) I would avoid using the word "impact" in cases where you do not establish causality or direction of effects (which is not established simply by temporal precedence), such as on line 137.

4) I think that the treatment of REMpre<30 s requires some more consideration/justification. In the text, the authors say that "we found that for values of REMpre < 150 s,142 ln(|N|) forms a bimodal distribution". But this is clearly not the case for REMpre < 30 s, so this statement is not accurate. The argument for effectively treating this as bimodal on Line 723 is not strong. A distribution can be unimodal without being Gaussian, as appears to be the case here. A decision on the basis of the Shapiro-Wilk test therefore does not make sense. My interpretation looking at Fig. 2E is that the unimodality for short REM bouts is effectively due to blending of the two types of NREM bouts (short and long), which would arguably justify treating it as a bimodal distribution.

5) Line 752: "We calculated the residual sum of squares (RSS) forlong753 both the linear and logarithmic fits and chose the function for which the RSS was lower." - This is a 2-parameter vs. 3-parameter fit comparison. Since the slope of the logarithmic function is a/(x+b)+c, it can also be made to approximate a linear fit for very large |b|. Without bounds on a, b, c, this does not seem to be a meaningful comparison (i.e., the log function is theoretically guaranteed to do better), as the logarithmic function could always be made to fit the data at least as well as the linear fit based on RSS. The comparison would make sense only with adjustment for the number of parameters (e.g., with AIC). It may be simplest to just drop the linear fits.

Reviewer #2: uploaded as an attachment

Reviewer #3: This manuscript is an important and stimulating contribution to the understanding of the ultradian organization of sleep that gives further support to notion that hourglass processes may significantly contribute to shape the architecture of the sleep –wake cycle. It also recognizes the centrality of REM sleep episodes (and its associated hourglass mechanism) in the timing of ultradian sleep cycles. The analytic strategy is elegant, solid and highly reliable, and is an invitation to be replicated in other animal models including humans. This work also confirms in mice the existence of two separate categories of REM sleep episodes (sequential and single) that were described by the Bologna group more than twenty years ago (Amici et al 1994 ). It also confirms that the factor “time of day” (see below) affects the timing of the REMs hourglass process (as described in ref 24: Vivaldi et al. 2005).

In my opinion, the main theoretical contribution the authors are: (i) incorporates the recently proposed REMs dependent “refractory period” as a relevant timing mechanism within the ultradian cycle of mice as has been described in rats; (ii) that the REMs “refractory period” is an exclusive attribute of single REMs episodes.; (iii) that REMs episodes duration may be determined within ultradian cycles by REMs propensity estimated by the Cumulative Density Function of NREM.

I have the following observations:

1. The definition of REM sleep propensity as the CDF is clear but the relationship between N an W in the buildup of REM sleep propensity is in my opinion not clearly stated. It is a complicated issue because it could be discussed form the point of view of the relationship between REMpre vs. inter-REMinterval, and between inter-REMinterval vs REMpost.

The paper discussed mostly the second alternative, as stated in line 653: “REM propensity and its impact on the REM episode duration i s more closely associated with the time spent i n NREM sleep than with the combined time i n NREM sleep and wake.” Authors limited the analysis of REMpre vs. inter-REMinterval to the results presented in Fig 1B. The problem of direct REMs episode duration with following amount of wakefulness (in inter-REM-interval) is the interference (masking) of feeding, drinking, exploring, etc, activities that obviously are outside sleep modulatory proceses, introducing loud noise in the timing of wakefulness with respect to the ultradian temporal frame.

I would suggest to explore the relationship of REMpre and REMpost with following interval duration and W duration by setting CDF constant.

2. Detailed information should be presented regarding the inter-individual variability. On the other hand, no details are presented with respect to the validation of automated sleep scoring (line 709: “we manually verified the automatic classification using a graphical user i nterface, visualizing the raw EEG, EMG signals, spectrograms and hypnogram”).

3. Results regarding night-day comparisons were not included in Discussion section? On the other hand I consider inappropriate the term “circadian” when referring to night-day comparisons. Circadian processes are estimated in circadian time or under specific experimental conditions that exclude local or zeitgeber time as main determinant, and is not the case in this experiment.

4. Presented results (particularly those related to sequential vs. single REMs) should be discussed in the light of evidences obtained in rats. In this sense I would suggest authors to review the following reference (Amici, et al. https://doi.org/10.1142/9781860947186_0008, in Parmeggiani &Velluti Eds. The Physiologic Nature of Sleep) where rat neurophysiological and vegetative aspects of REM sleep physiology are summarized. In this sense, there are interesting similarities between mice and rats in the neurophysiological description of sequential episodes.

Minor:

Place references 31 (Amici 1994) before reference 31 (Zamboni et al. 1999) , as the original description correspond of sequential/single episodes correspond to Amici.

Reviewer #4: Summary: This is a very interesting paper that uses a wide range of statistical approaches to provide a highly detailed analysis of REM sleep in mice. As the authors suggest, their analyses establish a useful framework for characterizing “the ultradian regulation of REM sleep in health and disease.” In addition, these analyses identify three major factors shaping ultradian regulation of sleep that have implications for the physiological basis of REM sleep regulation. My specific questions and concerns are listed below.

Major concerns:

1. p.4, line 94: more detail/explanation is needed to support the claim that REM sleep sequences have been observed in multiple species including humans. The timescales of these sequences (and ultradian cycles in general) vary widely across species, so these terms should be defined carefully.

2. P. 25: the authors describe a semi-automatic scoring approach for determining sleep/wake behavior. Has this scoring approach been published previously? Validated against other rodent scoring?

3. Related to the question about scoring, how was the 20 sec threshold for microarousals determined? This choice significantly affects the analysis through the definition of |N| and |W|.

4. The authors should be careful with the language used in the interpretation of the model, particularly in the last paragraph of the Results on P. 18 and 19. Since the model has been fit to the data, the model parameters describe the data but do not explain features such as an increased percentage of single cycles or contribute to an increase in NREM relative to REM sleep. Similarly, on p. 2, line 24: it seems too strong to say that the model “confirms” the existence of two types of sleep cycles; it would be better to say “supports” here.

Minor concerns:

1. Some of the citations in the Introduction seem a little sparse; e.g.,

a. P.3, line 67: citations to REM networks should include work from Yang Dan’s group such as Chen et al., Neuron, 2018

b. P. 4, line 79: citations to REM homeostat should include Franken et al. (ref [51])

2. P. 6, line 141: more detail is needed for the construction of the probability densities in Figure 2B; I am interpreting the terminology “REM_pre splits” to refer to subsets of REM_pre, but I am unclear on the significance of these subsets. How were the 30 second bins chosen? Were there similar numbers of REM_pre episodes in each subset? Approximately how many?

3. P. 8, line 198-201: the interpretation of the duration of the refractory period as a function of REM_pre needs clarification. Does this correspond to the increased black region below the red band in the heat map in Fig 2E?

4. P. 9, lines 216 – 223: this comparison with previous work may be better in the Discussion.

5. P. 9, line 229: what is the |N| duration associated with optimal separation in Figure 3A (right)? How does this threshold vary with |N|, and using this threshold, what percentage of sequential cycles are misclassified as long and vice versa (for different values of |N|)?

6. P. 9, line 235: the interpretation of a sequence of REM sleep with only one cycle as comprising 2 REM periods is given in the caption of Figure 3 caption but not in the text. It would be useful to define in both places.

7. P. 17, line 498: the authors claim that the REM duration is more closely correlated with the CDF than the preceding NREM duration. Is this conclusion supported by a formal analysis of the correlations (e.g., Williams correlation test)?

8. P 18, line 508: careful language should be used when talking about circadian differences; although circadian modulation of REM sleep has been established in other work, these results focus on light/dark differences.

9. P. 18, line 537-538: it would help to remind the readers that k_long is the weighting parameter for the long REM bouts in the Gaussian mixture model.

10. P. 20, line 583: the sleep phase N1 is increased in narcolepsy, and, although this light phase of sleep is associated with transitions to wake, fragmentation of N1 is not increased in narcolepsy (type 1) compared to controls (Maski et al., SLEEP, 2021).

**Have the authors made all data and (if applicable) computational code underlying the findings in their manuscript fully available?**

Reviewer #1: Yes

Reviewer #2: Yes

Reviewer #3: Yes

Reviewer #4: **No: **I may have missed it, but it looked like only the raw data and not the code were available in the repository the authors indicated.

PLOS authors have the option to publish the peer review history of their article (what does this mean?). If published, this will include your full peer review and any attached files.

Reviewer #1: No

Reviewer #2: **Yes: **H. Craig Heller

Reviewer #3: No

Reviewer #4: No

Figure Files:

Data Requirements:

Reproducibility:

References:

---

## [Decision Letter · Decision Letter 1]

21 Jul 2021

Dear Dr Weber,

Thank you very much for submitting your manuscript "A probabilistic model for the ultradian timing of REM sleep in mice" for consideration at PLOS Computational Biology. As with all papers reviewed by the journal, your manuscript was reviewed by members of the editorial board and by several independent reviewers. The reviewers appreciated the attention to an important topic. Based on the reviews, we are likely to accept this manuscript for publication, providing that you modify the manuscript according to the review recommendations.

Dear Franz:

Sorry for the delay in responding. Reviewer 4 still found a minor issue that needs your attention. Reviewer 2 (Craig Heller) maintains that your use of the term refractory is incorrect. I disagree with Craig as I introduced this term in the 2002 paper which you already cited and also the more recent work by Le Bon (2021) and Vivaldi (2020), both already cited, use this terminology (there are probably other publications). Even the algorithm that Joel used to determine NRTs (Benington, Kodali, Heller SLEEP 1994), and I used in the 2002 paper, has a 40s delay build-in, in which NRTs cannot occur. I leave it up to you whether you want to add a sentence qualifying your use of the term refractory. To conclude, consider your manuscript as provisionally accepted and I hope you can still revise your manuscript by addressing these last two issues.

with kind regards,

Paul

Sincerely,

Paul Franken

Guest Editor

PLOS Computational Biology

Wolfgang Einhäuser

Deputy Editor

PLOS Computational Biology

[LINK]

Reviewer's Responses to Questions

**Comments to the Authors:**

Reviewer #1: Thank you for carefully addressing all my comments. This is an excellent paper and a significant contribution to our understanding of REM/NREM sleep cycles.

Reviewer #2: Thanks for the interesting discussion. I re-iterate that I am not scoring your manuscript even though I think your data are great. I just don't have the expertise to evaluate your modeling that is a major portion of the paper. However, I was trying to convince you that you have a perfectly good physiological explanation of your data without invoking a mysterious refractory phase, which the data do not support. That explanation is classical homeostatic regulation, which your data do support. The definition of refractory is to be insensitive to a stimulus that was previously effective. A neuron is refractory to depolarizing input immediately following an action potential. A man is refractory to sexual stimulation following orgasm. If you went out into the cold without a coat and started to shiver, but then put on a coat and stopped shivering, are you refractory to cold? No, you have just eliminated the stimulus. At one point you also seem to accept someone else's description of the REM/NREM process as being like an hour glass --- no it is not. Maybe an old fashioned ice box is like an hour glass as the ice melts, but a modern refrigerator is a regulated system. The NREM REM cycles are regulated homeostatically as shown by both our results. If you are looking for a function of REM, you are more likely to find it by looking for the feedback information relating NREM to REM, and not in a supposed mechanism of refractoriness.

As we showed, the brief REM episodes are likely to occur at any time in the classical sleep cycle, but are increasingly likely as the cycle continues. Also, a brief REM episode is likely to be followed soon by another one, and so on with increasing rapidity. These brief REM episodes are failures to maintain REM, and when they fail they do not dissipate much REM propensity, so they come again sooner and sooner until a REM bout is sustained and dissipates the REM need.

Physiologically, we need to be looking for what process is exhausted or built up during the hyperpolarized NREM state that has to be reversed during REM. That is investigating a homeostatic process and not a refractory mechanism.

You can interpret your data anyway you want in your paper, of course. But, I just want you to try thinking of your beautiful data in a physiological way.

Reviewer #3: I have no further concerns.

Reviewer #4: Summary: This is a very interesting paper that uses a wide range of statistical approaches to provide a highly detailed analysis of REM sleep and the impact of REM sleep on NREM sleep in mice. The authors have addressed my previous comments in the revised manuscript, and the new Discussion subsection on Sequential Sleep Cycles is a great addition. One few minor suggestion:

Lines 612-615: The authors state that “In rats and, as shown here, in mice the average duration of REM sleep episodes preceding a sequential cycle is on average shorter than that before a single cycle, and the frequency of sequential cycles is reduced during the dark period [33].” Is this comment actually restricted to the REM sleep episodes preceding these events or should it describe REM sleep episodes within sequential cycles compared to REM sleep episodes in a single cycle? If the authors are highlighting a feature of the first REM sleep episode in a sequential cycle, it would be helpful to clarify this point.

**Have the authors made all data and (if applicable) computational code underlying the findings in their manuscript fully available?**

Reviewer #1: Yes

Reviewer #2: Yes

Reviewer #3: Yes

Reviewer #4: Yes

PLOS authors have the option to publish the peer review history of their article (what does this mean?). If published, this will include your full peer review and any attached files.

Reviewer #1: **Yes: **Andrew J K Phillips

Reviewer #2: **Yes: **H. Craig Heller

Reviewer #3: **Yes: **Adrián Ocampo-Garcés, MD, PhD

Laboratorio de Sueño y Cronobiología

ICBM, Facultad de Medicina

Universidad de Chile, Santiago

Chile

Reviewer #4: **Yes: **Cecilia Diniz Behn

Figure Files:

Data Requirements:

Reproducibility:

References:

---

## [Editor Report · Decision Letter 2]

29 Jul 2021

Dear Dr Weber,

We are pleased to inform you that your manuscript 'A probabilistic model for the ultradian timing of REM sleep in mice' has been provisionally accepted for publication in PLOS Computational Biology.

Best regards,

Paul Franken

Guest Editor

PLOS Computational Biology

Wolfgang Einhäuser

Deputy Editor

PLOS Computational Biology

---

## [Editor Report · Acceptance letter]

20 Aug 2021

PCOMPBIOL-D-21-00423R2 

A probabilistic model for the ultradian timing of REM sleep in mice

Dear Dr Weber,

I am pleased to inform you that your manuscript has been formally accepted for publication in PLOS Computational Biology. Your manuscript is now with our production department and you will be notified of the publication date in due course.

With kind regards,

Livia Horvath
